# MANAR: Memory-augmented Attention with Navigational Abstract Conceptual Representation

## Abstract

We introduce **MANAR**, a linear-time attention layer for transformer *encoders* that can directly inherit weights from a pretrained Transformer's multi-head attention (MHA) — a property that distinguishes it from existing linear-time alternatives such as Mamba, RetNet, and Linear Attention, which require training from scratch and therefore forfeit access to the representational capital accumulated in large pretrained Transformers. MANAR augments MHA with a trainable external memory and a constant-size **Abstract Conceptual Representation (ACR)**, a design inspired by the global-workspace bottleneck described in cognitive models of perception. The architecture follows a two-stage logic: (i) an *integration phase*, in which retrieved memory concepts are combined with the input sequence to form the ACR, a compact global state of the input; and (ii) a *broadcasting phase*, in which the ACR informs the contextualization of each token together with a local context window, replacing all-to-all attention. Routing global information through a constant-sized ACR yields strictly linear time and memory complexity *when the local context window is held to a constant independent of sequence length,* a regime we verify empirically (single-layer latency grows linearly in $n$, $R^2$=0.998; Sec. 4.7). Because MANAR preserves the semantic roles of the standard MHA projections, knowledge transfer from pretrained transformers reduces to a direct weight-copy, and we show that transferred models recover and then exceed the accuracy of their sources at a fraction of the from-scratch training budget. As a diagnostic of where this capacity comes from, we show the memory pathway lets outputs leave the convex hull of the input value vectors (which softmax attention cannot), confirming the pathway is active even on a frozen backbone; we do not claim this geometry by itself causes higher accuracy. Across language, vision, and speech, MANAR is competitive with strong baselines (GLUE 84.3, ImageNet-1K 83.9% top-1, LibriSpeech 2.7%/6.4% WER). MANAR delivers large single-layer latency and peak-memory reductions versus a standard (non-FlashAttention) MHA — e.g. a 23× peak-memory reduction and ~15× lower latency at 8,192 tokens, beyond which standard MHA exceeds GPU memory. Against the optimized FlashAttention-2 kernel in bf16, MANAR's advantage instead emerges in the long-context regime (it overtakes FlashAttention-2 at $n \gtrsim$ 8K and is $\sim$ 5.5× faster at 32K), the setting where its linear scaling matters most. We further validate long-context *quality*: transferring from strong pretrained long-context encoders, MANAR reaches 88.1% on arXiv long-document classification at 4,096 tokens (matching the full Longformer) and stays within half a point of full-attention ModernBERT at 8,192 tokens (with RoPE).

## 1 Introduction

Since its introduction, the transformer architecture (Vaswani et al., 2017) has achieved remarkable success across a wide spectrum of domains, including natural language processing (Vaswani et al., 2017; Karpukhin et al., 2020; Touvron et al., 2023; Liu et al., 2024a; Warner et al., 2024), computer vision (Dosovitskiy et al., 2020; Arnab et al., 2021; Shehzadi et al., 2023), speech recognition (Schneider et al., 2019; Baevski et al., 2022; Liu et al., 2023), bioinformatics (Brandes et al., 2022; Acera Mateos et al., 2021; Jahshan & Yavits, 2024), and many other domains. At the heart of this success lies the attention mechanism, which enables

every token to attend to all other tokens in a sequence, yielding highly expressive, sequence-wide contextualizations and facilitating efficient parallel training. However, the same all-to-all contextualization that powers the expressivity of multi-head attention (MHA) introduces significant scalability bottlenecks. The quadratic time and memory complexity with respect to sequence length, together with the need to store a linearly growing, unbounded context in autoregressive generation, limits both the efficiency and the reach of contemporary attention-based models. This challenge has become increasingly pressing as workloads shift toward longer contexts, higher-resolution inputs, and larger-scale models, motivating numerous architectural and algorithmic strategies to address these limitations, including hierarchical attention, recurrence, compression, and explicit memory mechanisms.

Among the most widely adopted approaches are quantization and knowledge distillation, which have succeeded in compressing the memory and compute footprint of transformer models. Quantization (Ashkboos et al., 2024; Liu et al., 2024c;d; Xiao et al., 2023) enables more efficient computations and reduced memory footprint by lowering precision, sometimes as low as 4-bit per element (Liu et al., 2024c). Distillation (Bing et al., 2025; Han et al., 2024; Mukherjee et al., 2021) transfers knowledge from large models to smaller ones. Despite their practical utility, these methods fall short of solving the core issue: the direct all-to-all token contextualization is preserved, so context size remains unbounded and computational complexity quadratic.

To address the unbounded-context problem, several lines of work introduce recurrence, compression, or explicit memory to extend context length without quadratic growth. Transformer-XL (Dai et al., 2019) introduces segment-level recurrence, allowing hidden states from previous segments to be reused as extended context while avoiding full recomputation. Building on this idea, the Compressive Transformer (Rae et al., 2019) augments recurrence with a compressed long-term memory, enabling retention of distant context at reduced resolution. Memory Transformer (Burtsev et al., 2020) augments standard transformers by adding trainable memory tokens that accumulate non-local representations and help capture properties of the entire sequence beyond what element-wise contextual embeddings provide. Such memory tokens can act as dedicated slots for storing global patterns, and bottlenecks can restrict global information propagation to emphasize essential representations.

More recently, many works (Xiao et al., 2024; Fountas et al., 2025; Sun et al., 2024; Lee et al., 2024; Liu et al., 2024b; Mohtashami & Jaggi, 2023; Wu et al., 2022) focus on offloading the KV cache into lower memory tiers (e.g., CPU DRAM) and sparsely selecting KV pairs to attend to. These techniques exploit the empirical observation that, during contextualization, only a small subset of tokens significantly contributes due to highly peaked attention distributions. To capitalize on this behavior, several approaches augment multi-head attention with explicit memory units that store contextual blocks (i.e., KV blocks) for later retrieval. For example, InfLLM (Xiao et al., 2024) contextualizes each token using both a local context window and a set of highly related KV blocks retrieved via approximate nearest-neighbor search. The retrieved blocks and the local context jointly participate in token contextualization, enabling scalable long-context inference. However, while these strategies effectively reduce active computation time, they do not resolve the fundamental architectural burden of maintaining a linearly growing and unbounded KV cache, which remains a primary bottleneck for scaling to ultra-long sequences.

Many works seek to completely replace the standard attention mechanism with alternative architectures designed for enhanced scalability. For example, Titans (Behrouz et al., 2024) and ATLAS (Behrouz et al., 2025) augment standard attention (used as short-term memory) with explicit long-term neural memory modules, which are updated using test-time training and optional persistent memory to retrieve and fuse past information alongside current-context attention for scalable long-range modeling. State space models (Gu et al., 2021) formulate sequence modeling as linear dynamical systems, allowing fast, parallel computation and effective modeling of long-range context. Mamba (Gu & Dao, 2023) extends this paradigm by introducing selective sequence modeling through dynamic state selection and gating, enabling efficient and expressive representations for extremely long inputs (Dao & Gu, 2024). Other notable advances, such as RetNet (Sun et al., 2023), propose recurrent architectures with retention mechanisms that further boost performance, demonstrating strong results on long-context benchmarks. These approaches fundamentally rethink sequential model design and show particular promise in ultra-long sequence regimes.

A separate line of work introduces a fixed-size set of latent vectors that mediates between inputs and outputs, structurally close to the design we adopt. Set Transformer (Lee et al., 2019) introduces inducing-point cross-attention for permutation-invariant set processing, reducing the input–input attention to input–inducing-point attention with constant-size latents. Perceiver (Jaegle et al., 2021) and Perceiver IO (Jaegle et al., 2022) extend this idea to general perception, compressing arbitrarily long inputs into a constant-size latent array and broadcasting the latents back to outputs through cross-attention. Multi-head Latent Attention (MLA), introduced in DeepSeek-V2 (DeepSeek-AI, 2024), applies low-rank K/V projections within standard attention to reduce KV-cache size, but remains an $O(n^2)$ within-attention modification. MANAR shares the constant-size-bottleneck philosophy of Set Transformer and Perceiver IO, but differs from them in two ways relevant to practitioners: (i) MANAR's projections are a strict re-parameterization of standard MHA's $Q$, $K$, $V$, and output projections, so weights from a pretrained Transformer can be transferred by direct copy — a property that Perceiver IO's latent-array design does not possess; and (ii) MANAR augments the bottleneck with a retrievable external concept memory, decoupling the size of the latent state (the ACR) from the size of the long-term knowledge store.

Nonetheless, a significant practical drawback of these alternative linear-time architectures is that they fundamentally alter the parameterization of the contextualization mechanism. Because they replace the standard attention mechanism with structurally different formulations, such models cannot directly inherit or easily transfer knowledge from existing pretrained transformer attention weights. This structural incompatibility creates a substantial barrier to practical adoption, as it prevents these architectures from leveraging the vast representational knowledge already stored in large-scale pretrained models.

We introduce MANAR [1], a memory-centric attention architecture that functions as a contextualization layer and can be plugged into commodity transformer encoder models. The architecture is inspired by cognitive processes in which perception and comprehension depend not only on sensory inputs but also on internalized concepts built from prior experience. When presented with an external input, the brain builds a mental image based on memorized concepts associated with the observed input and their relationships; this mental image then guides contextualization, the process in which meaning is assigned to each observed input occurrence.

Concretely, given an input sequence, MANAR (i) retrieves memory concepts and constructs a full-context, constant-sized Abstract Conceptual Representation that functions as a mental image of the sequence's global themes, and (ii) contextualizes each input token using this ACR together with a local context window, avoiding direct all-to-all contextualization. MANAR can be integrated as a drop-in replacement for standard MHA layers, making it practically deployable in existing transformer encoder stacks; weight-copy from pretrained models enables rapid adaptation by training only the additional memory-related parameters, and the design supports application areas such as information retrieval, knowledge management, and data or text mining. Empirical evaluation across language understanding, image classification, and speech recognition shows that MANAR is competitive with strong baselines while delivering substantial inference speedups and peak GPU memory reductions as sequence length increases.

The MANAR architecture is inspired by Global Workspace Theory (GWT) (Baars, 1988), which hypothesizes that the brain functions through a central workspace where information from various specialized modules is integrated and subsequently "broadcast" to the rest of the system (Baars, 2002). While standard multi-head attention (MHA) allows for all-to-all communication, it lacks the functional bottleneck that GWT suggests is necessary for coherent global integration (Dehaene et al., 1998). We use the GWT framing as an interpretive guide for the architectural choice of routing global information through a constant-size Abstract Conceptual Representation (ACR), not as a claim that MANAR mechanistically realizes a cognitive theory.

## 2  Preliminaries and background

We begin by formalizing the notions of *concept* and *contextualization*, which we use throughout this work to reason about attention mechanisms and to motivate the MANAR architecture. These notions are inspired by cognitive models of human perception as well as by the operational structure of modern attention-based neural networks.

---

[1]Code: xxxxxxxxxxxxxxxxxxxxxxxxxx

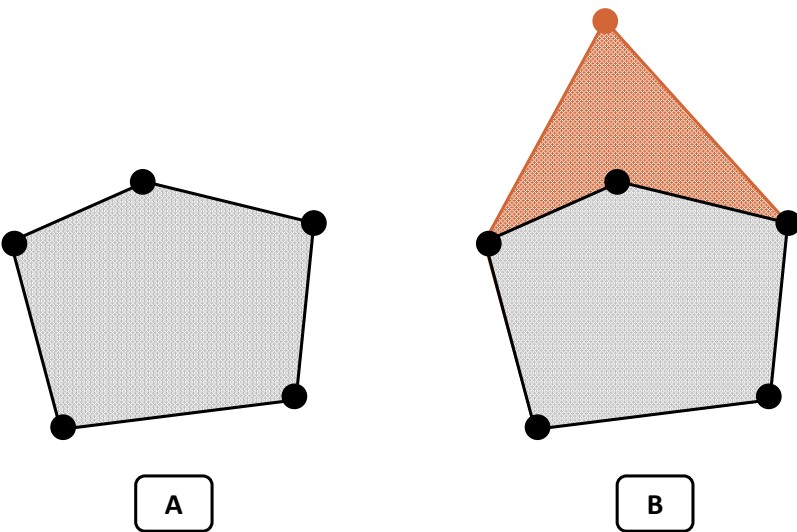

Figure 1: Geometric interpretation of contextualization with and without retrieved memory concepts.

**Concept**   We model the fundamental unit of information processed by attention layers as a *concept*. A concept encapsulates not only semantic content but also the mechanisms by which this content interacts with other information. Concretely, a concept is represented as a triplet $(q, k, v)$, where the *query q* determines how the concept seeks relevant context, the *key k* determines how it contributes to the contextualization of other concepts, and the *value v* represents the semantic content carried by the concept. Following the GWT analogy, individual tokens and memorized concepts can be viewed as specialized resources competing for representation within the global workspace (Dehaene & Naccache, 2001); the ACR plays the role of a compact, constant-sized representation through which global information is integrated before influencing the wider network.

**Contextualization**   Contextualization is the process by which the meaning of a concept is refined through interaction with other concepts. Given a concept $x = (q, k, v)$ and a set of concepts $C = \{(c_1^q, c_1^k, c_1^v), \ldots, (c_n^q, c_n^k, c_n^v)\}$, contextualization produces an updated representation by aggregating the values of concepts in $C$ according to their relevance to $x$. Relevance is measured via a similarity function between the query of $x$ and the keys of the contextualizing concepts.

Formally, the contextualized representation $y$ of $x$ with respect to $C$ is given by

$$y = \sum_{(c^q, c^k, c^v) \in \tilde{C}} S(q, c^k) \, c^v, \quad \text{where } \tilde{C} = C \cup \{x\},$$

with $S(\cdot, \cdot) \geq 0$ and $\sum_{(c^q, c^k, c^v) \in \tilde{C}} S(q, c^k) = 1$.

When the similarity weights are non-negative and normalized (as in standard softmax attention), contextualization produces a convex combination of value vectors. Consequently, the contextualized representation is constrained to lie within the convex hull spanned by the values of the contextualizing concepts. This geometric interpretation highlights a property of standard contextualization mechanisms: when contextualization is restricted to the input sequence alone, the expressivity of the output is bounded by the convex hull of the input token values, as illustrated in Fig. 1(A).

When contextualization incorporates *retrieved memorized concepts* that are not present in the input sequence, this constraint is relaxed: as shown in Fig. 1(B), the inclusion of external concept values expands the reachable region in representation space. We refer to this capability as *non-convex contextualization*, in the sense that the output can be expressed as a convex combination of input values *plus* retrieved memory values, but not (in general) as a convex combination of input values alone.

**Multi-head attention as contextualization** The multi-head attention (MHA) mechanism can be interpreted as a particular instantiation of the contextualization process described above, where each input token is conceptualized by a learned $(q, k, v)$ triplet and contextualized exclusively by other input tokens in the sequence.

In this work, we develop a different contextualization process. When an external sequence of inputs is presented, internal memorized concepts that are strongly associated with the input are retrieved. Links and associations are then formed between input tokens and retrieved memorized concepts. These links take the shape of a constructed global conceptual representation of the perceived input. This constructed representation then guides the contextualization of the presented inputs. We can examine human reading comprehension as an analogous example. When presented with a text, the words stimulate the brain to retrieve memorized concepts associated with them. Throughout the reading process, the brain constructs an abstract mental representation capturing the perceived meaning of what is being read and its connections to memory. Each word is then contextualized in light of this constructed abstract representation (Kewenig et al., 2024; Keller et al., 2024).

## 3 MANAR

MANAR contextualizes input tokens by neighboring tokens as well as by retrieved memorized concepts not present in the input sequence. To make this possible, MANAR integrates a memory unit that retains memorized concepts. When an input sequence of tokens $X$ is presented, $m$ search patterns are generated and applied to retrieve $m$ memorized concepts from the memory unit in a fast and scalable manner. After retrieval, these internal memory concepts are linked and associated to the presented input forming the Abstract Conceptual Representation ($ACR$). Then, input tokens are contextualized by the $ACR$ as well as the local context window of the token to formulate the output of the layer.

### 3.1 Notations

We start by discussing some notations that are used consistently throughout the paper. Let $X \in \mathbb{R}^{n \times D}$ represent the input sequence of tokens, where $n$ is the sequence length and $D$ is the dimension of each observed input token. Let $\mathcal{M}_i$ be the $i$-th retrieved memory concept which is consistently represented as a qkv-tuple $\mathcal{M}_i = (c_i^q, c_i^k, c_i^v)$. Use $m$ to refer to the number of retrieved memory concepts, and $M$ to refer to the total memory size (i.e., the total number of memory cells in the memory unit). The $ACR$ size is $m \times d$ (where $d$ is the per-head dimension), a row for each retrieved memory concept. We refer to the $i$-th row of the $ACR$ by $ACR_i$ or $r_i$ interchangeably. We refer to $d$ as the intra-layer dimension (i.e., per-head dimension), and we assume $D = hd$ where $h$ is the number of heads. Lastly, the $i$-th output (i.e., the contextualized $i$-th token) is referred to as $y_i$. In this work we follow a row-major representation.

### 3.2 ACR construction and Token Contextualization

Throughout Sec. 3.2 we assume the existence of $m$ retrieved memory concepts $\left\{\mathcal{M}_i = (c_i^q, c_i^k, c_i^v)\right\}_{i=1}^m$ decoupling the process of memory retrieval from the rest of the MANAR architecture. Memory retrieval is discussed separately in Sec. 3.3. Calculations are made considering a single-head architecture. Multi-head architecture generalization is made in Section 3.4.

MANAR defines four learnable projection matrices $W_q, W_k^{\mathcal{M}}, W_k, W_v \in \mathbb{R}^{D \times d}$ corresponding to the token's query, ACR key, contextualization key, and value, respectively. $W_k^r \in \mathbb{R}^{d \times d}$ represents the projection responsible for converting $ACR$'s into "token contextualization" key-space. Moreover, we define the Region of Interest of an index $i$ with a neighborhood $l$ as:

$$ROI^l(i) = \{j : \max(0, i - l + 1) < j \leq \min(n, i + l)\}$$

where $ROI^l(i)[j]$ represents the $j$-th smallest element in the set. The $ROI$ is used to represent the local context window in the process of token contextualization. We refer to the local context window length as the maximal size of $ROI^l$, which is $2l$. Equipped with these learnable parameters and the $ROI$, the logic of the

MANAR layer operating in two stages (i.e., the Integration and Broadcasting stages), the ACR construction and the token contextualization, is defined as follows:

- Conceptualization:

$$k_i^{\mathcal{M}} = x_i W_k^{\mathcal{M}} \qquad\qquad q_i = x_i W_q \qquad (1)$$
$$k_i = x_i W_k \qquad\qquad v_i = x_i W_v \qquad (2)$$

- Integration stage (ACR Construction):

$$r_i = S_{i,0} c_i^v + \sum_{j=1}^{n} S_{i,j} v_j \qquad (3)$$

where

$$(S_{i,0}, \ldots, S_{i,n}) = \mathrm{softmax}\Big( c_i^q (c_i^k)^T / \sqrt{d}, $$
$$c_i^q (k_1^{\mathcal{M}})^T / \sqrt{d}, \ldots, c_i^q (k_n^{\mathcal{M}})^T / \sqrt{d} \Big).$$

The process of ACR construction can be seen as contextualizing each retrieved memory concept by the conceptualized input tokens. Concretely, the $i$-th ACR vector, $r_i$, represents the meaning of the memorized concept shifted according to how strongly that memory concept associates with each observed token. The strength of an association is measured by the inner product $c_i^q \cdot (k_j^{\mathcal{M}})^T$. Intuitively, this mirrors the cognitive analogy: a mental image of a situation blends internal memorized concepts with incoming evidence, weighted by the perceived relevance of each piece of evidence to those concepts.

- Broadcasting stage (Token Contextualization):

$$y_i = \underbrace{\sum_{j=1}^{m} \hat{S}_{i,j} r_j}_{\text{global}} + \underbrace{\sum_{j=1}^{L} \tilde{S}_{i,j} v_{ROI^l(i)[j]}}_{\text{local}} \qquad (4)$$

where

$$(\hat{S}_1, \ldots, \hat{S}_m, \tilde{S}_1, \ldots, \tilde{S}_L) =$$
$$\mathrm{softmax}\Big( q_i (r_1 W_k^r)^T / \sqrt{d}, \ldots, q_i (r_m W_k^r)^T / \sqrt{d},$$
$$q_i k_{ROI^l(i)[1]}^T / \sqrt{d}, \ldots, q_i k_{ROI^l(i)[L]}^T / \sqrt{d} \Big)$$

$$L = |ROI^l(i)|.$$

Navigated by the *ACR*, MANAR contextualizes each input token in light of this global representation as well as all the input information that could be perceived at once (i.e., the local context window), represented by the *ROI* of the token. Hence, the meaning each contextualized token holds is influenced both by association with the *ACR* and with neighboring tokens.

Since the ACR is constructed around memorized concepts and the meaning they hold, MANAR - contextualized tokens can take values not bounded by the meaning space spanned by the convex hull of inputs $\{x W_v : x = X_i, 1 \le i \le n\}$. We refer to this behavior as *non-convex contextualization* (with respect

to the input value vectors). This effect is expressed by the following derivation of $r_i$:

$$
\begin{aligned}
r_i &= S_{i,0} c_i^v + (1 - S_{i,0}) \sum_{j=1}^{n} \frac{S_{i,j}}{1 - S_{i,0}} v_j \\
&= S_{i,0} c_i^v + (1 - S_{i,0}) \\
&\quad \times \sum_{j=1}^{n} \frac{e^{c_i^q (k_j^{\mathcal{M}})^T} / \left( e^{c_i^q (c_i^k)^T} + \sum_{l=1}^{n} e^{c_i^q (k_l^{\mathcal{M}})^T} \right)}{\sum_{l=1}^{n} e^{c_i^q (k_l^{\mathcal{M}})^T} / \left( e^{c_i^q (c_i^k)^T} + \sum_{l=1}^{n} e^{c_i^q (k_l^{\mathcal{M}})^T} \right)} v_j \\
&= S_{i,0} \underbrace{c_i^v}_{B} + (1 - S_{i,0}) \underbrace{\sum_{j=1}^{n} \frac{e^{c_i^q (k_j^{\mathcal{M}})^T}}{\sum_{l=1}^{n} e^{c_i^q (k_l^{\mathcal{M}})^T}} v_j}_{A}
\end{aligned}
\tag{5}
$$

Eq. 5 demonstrates that each ACR row $r_i$ is a weighted sum of two terms: (A) an expression having the same form as the output of MHA; (B) a correction term that can shift $r_i$ outside the convex hull of the input token values, as illustrated in Fig. 1(B).

The two-stage logic of MANAR aligns with the high-level mechanics of a global workspace. Stage 1 (ACR Construction) plays the role of an integration phase, in which relevant memorized concepts are retrieved and shifted based on their association with the input sequence to form a compact global state. Stage 2 (Token Contextualization) plays the role of a broadcasting phase, in which the global state maintained in the ACR informs the contextualization of each individual token, so that local perception is interpreted in light of global context (Baars, 2002).

### 3.3 The Memory Unit

In this section, we discuss the memory unit and the memory retrieval process of $m$ concepts, $\mathcal{M}_i = (c_i^q, c_i^k, c_i^v)$, completing the full picture of MANAR .

The memory unit contains $M$ memory cells. Each memory cell retains a concept, $\boldsymbol{\mu}_i = (\mu_i^q, \mu_i^k, \mu_i^v)$, where $0 < i \leq M$. The memory retrieval process involves the creation of $m$ different search patterns as a function of input tokens. For each search pattern, top-$k$ memory cells are chosen on the basis of their similarity to the search pattern. The memory concept is calculated as a weighted sum of the contents of these matching top-$k$ memory cells. The logic of producing the search pattern is first formalized, then we detail how each search pattern drives retrieval.

To perform $m$ memory lookups, the model first constructs $m$ search patterns from the input sequence $X \in \mathbb{R}^{n \times D}$. Concretely, the model introduces $m$ learnable "mixer" vectors $mixer_i \in \mathbb{R}^d$ that aggregate information from tokens via a cross-attention operation where the queries are the mixer vectors and the keys/values come from the tokens. Let $W_k^{SP}, W_v^{SP} \in \mathbb{R}^{d \times d}$ be learnable projections; the $i$-th search pattern is then:

$$
\sigma_i = \text{softmax}\left( \frac{mixer_i \cdot (X W_k^{SP})^T}{\sqrt{d}} \right) \cdot X W_v^{SP}
\tag{6}
$$

Given a search pattern $\sigma_i$, the memory unit keys, a table of keys, one per each memory cell, $\xi \in \mathbb{R}^{M \times d}$, and the memory cells $\boldsymbol{\mu} \in \mathbb{R}^{M \times 3d}$, retrieval computes a soft combination of cells weighted by their similarity to $\sigma_i$. The retrieval step is:

$$
\begin{aligned}
I &= \text{SelectTopkIndices}(\sigma_i \cdot \xi^T); \\
s &= \text{softmax}(\sigma_i \cdot (\xi_I)^T); \\
\mathcal{M}_i &= s \cdot \boldsymbol{\mu}_I.
\end{aligned}
\tag{7}
$$

where $I$ is a set of indices, $s \in \mathbb{R}^k$, $\xi_I \in \mathbb{R}^{k \times d}$, $\boldsymbol{\mu}_I \in \mathbb{R}^{k \times (3d)}$, and the output $\mathcal{M}_i \in \mathbb{R}^{3d}$.

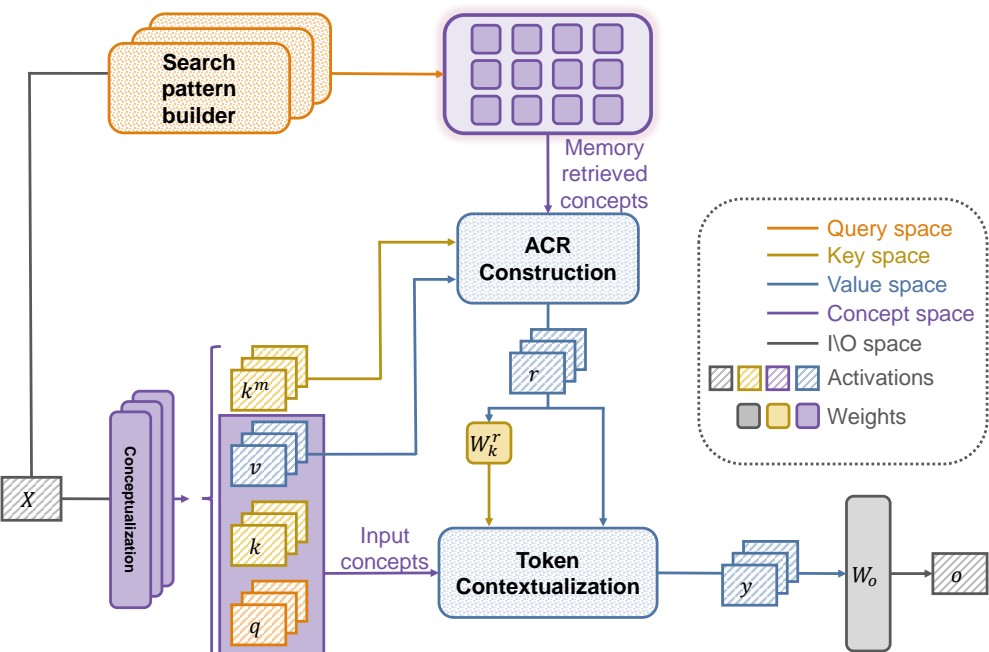

Figure 2: High level architecture of MANAR .

Increasing the memory size, $M$, makes naive nearest-neighbor scoring over all keys $\mathcal{O}(M)$ per search pattern prohibitive; fast approximate similarity search techniques could be used here (Johnson et al., 2019), but incorporating them is challenging when keys are continually trained and re-indexed. MANAR uses trainable product keys (Lample et al., 2019), which factor the key space into two tables $\xi^{(1)}, \xi^{(2)} \in \mathbb{R}^{\sqrt{M} \times \frac{d}{2}}$ whose (implicit) Cartesian product spans $M$ composite keys without materializing them. For lookup, split the search pattern as $\sigma_i = \left[\sigma_i^{(1)}; \sigma_i^{(2)}\right]$ with $\sigma_i^{(1)}, \sigma_i^{(2)} \in \mathbb{R}^{\frac{d}{2}}$, retrieve top-k indices and scores $(I_1, s_1)$ from $\xi^{(1)}$ and $(I_2, s_2)$ from $\xi^{(2)}$, then combine candidates by maximizing summed scores over pairs:

$$\arg \max_{j_1 \in I_1, \, j_2 \in I_2} s_1[j_1] + s_2[j_2]$$

which yields an efficient approximation to top-$k$ over the full $M$ composite keys while searching only the $\sqrt{M}$-sized half-key tables.

### 3.4 Multi-head architecture

To form a $h$-head MANAR layer we replicate the entire single-head conceptualization pipeline, including the search-pattern builder, one per head. Each head therefore learns its own token projections and search patterns, yet all heads reuse the same token-contextualization key projection $W_k^r$ and access one shared external memory. After each head completes retrieval, ACR construction and token contextualization, their outputs are concatenated and mapped back to $D$ dimensions through a single output projection, preserving the standard transformer interface.

## 4 Evaluation

To evaluate MANAR accuracy, performance, and memory usage, we apply it as a drop-in replacement to MHA in several transformer-based models, corresponding to language, image, and speech modalities. The chosen benchmarks are representative of application settings: e.g. search and document analysis, visual recognition, and speech interfaces.

Across all modalities, our primary goal is to isolate the effect of the *contextualization mechanism*. Unless explicitly stated otherwise, our comparisons follow the same contract: we keep the backbone architecture, training pipeline, and evaluation procedure fixed, and change only the contextualization layer (MHA → MANAR ). All models we compare are parameter-budget matched (all base-sized models are kept below 150M parameters). We report wall-clock inference comparisons in Sec. 4.7 (Performance Comparison).

Because training recipes are highly non-unique and domain-specific, we do not claim that every baseline is fully optimized for best possible accuracy under all settings. Instead, we focus on controlled, apples-to-apples runs where MANAR and the baseline share the same data, preprocessing, optimizer, schedule, and evaluation protocol, so that observed differences can be attributed to the contextualization mechanism.

When comparing vanilla transformer encoder architecture to MANAR -enabled one, we leave all other layers unmodified. In this section, we refer to any MANAR -enabled transformer encoder architecture, having a memory of size $M$ and $ACR$ of size $m$ that aggregates top-8 memory cells to assemble a memory concept, with a context window length of $C$ as `MANAR-M.m.C`.

## 4.1 Language Modeling

We evaluate MANAR on natural language understanding via masked language model (MLM) pre-training followed by fine-tuning on downstream GLUE tasks. The purpose of this experiment is to isolate the impact of replacing multi-head attention (MHA) with MANAR in a BERT-base–style encoder, under a controlled RoBERTa-like (Liu et al., 2019) training procedure and using absolute positional embeddings. We exclude Rotary Positional Embeddings (RoPE) in this section to avoid conflating architectural effects with positional encoding choices.

Following the controlled protocol of Sec. 4, we replace only MHA with MANAR while keeping all other encoder components and the training pipeline identical.

For controlled comparisons, we train two families of models from scratch on the English subset of C4 using the MosaicML (Portes et al., 2023) framework, following the RoBERTa recipe: Next Sentence Prediction is removed, dynamic MLM masking is applied with a 15% masking probability, the sequence length is 512 tokens, training runs for 30K steps (i.e., only a portion of the C4 English subset accounting for 135 GB of data) with a global batch size of 4K, and all models use the `bert-base-uncased` tokenizer (30,522 tokens). Optimization uses AdamW with peak learning rate $5 \times 10^{-4}$ and the linear warmup with linear decay learning schedule (Devlin et al., 2019; Baevski et al., 2022; Liu et al., 2019; Portes et al., 2023). All trained models use the same data subset and identical optimization and schedule settings. Within this controlled regime, we train (i) a RoBERTa baseline, and (ii) two MANAR variants that act as strict drop-in replacements for MHA while keeping all other encoder components unchanged.

In addition to these controlled runs, we also report the published GLUE results of BERT (Devlin et al., 2019) and data2vec (Baevski et al., 2022) in Table 1. We include BERT as a historically standard BERT-base reference point (trained under earlier, smaller-data regimes) and data2vec as a strong modern self-supervised baseline that achieves competitive GLUE performance. Importantly, these two rows are taken *as reported* and are not retrained; the controlled, apples-to-apples comparison in our setting is therefore between RoBERTa run and MANAR runs.

The first variant, `MANAR-484.64.128`, uses a memory of 484 concepts, retrieves $m = 64$ concepts to form the ACR, and uses a local context window length of $C = 128$. The larger-memory configuration, `MANAR-16K.128.128`[†] uses a hybrid architecture: every third encoder layer uses a memory size of 16K concepts, retrieving $m = 128$ concepts to form the ACR, while the remaining layers use an identical configuration as of the smaller configuration (i.e., $M = 484$, $m = 64$, and $C = 128$). This hybrid design increases memory capacity while keeping overall compute and parameters in the BERT-base scale (i.e. <150M parameters).

Table 1 reports GLUE results after fine-tuning. Under the controlled training regime (RoBERTa vs. MANAR variants), MANAR is competitive with the baseline and attains the strongest average score (84.3) among the models trained in this setting. Increasing memory capacity from `MANAR-484.64.128` to `MANAR-16K.128.128`[†] raises the average from 83.3 to 84.3, improving the majority of tasks (most notably CoLA and QQP) while

Table 1: GLUE development set results (%). Higher is better. MANAR MRPC, RTE, STS-B, CoLA, and SST are reported as mean $\pm$ std over 3–5 fine-tuning seeds; MNLI, QNLI, and QQP are single-seed (seed 19). Reference rows are as-reported / single-seed.

| Model | MNLI | QNLI | RTE | MRPC | QQP | STS-B | CoLA | SST | Avg. |
|---|---|---|---|---|---|---|---|---|---|
| *Reference baselines (as reported, not retrained)* | | | | | | | | | |
| BERT (Devlin et al., 2019)[‡] | 84.0/84.4 | 89.0 | 61.0 | 86.3 | 89.1 | 89.5 | 57.3 | 93.0 | 80.7 |
| data2vec (Baevski et al., 2022)[‡] | 83.2/83.0 | 90.9 | 67.0 | 90.2 | 89.1 | 87.2 | **62.2** | 91.8 | 82.7 |
| *Controlled runs, trained from scratch (this work, identical recipe)* | | | | | | | | | |
| RoBERTa (Liu et al., 2019) | **84.4**/84.1 | 90.4 | **75.3** | 89.8 | 89.6 | 89.3 | 57.4 | 92.3 | 83.4 |
| MANAR-484.64.128 | 83.7/83.4 | **91.3** | 71.4 $\pm$0.5 | 90.0 $\pm$0.5 | 89.1 | **90.1** $\pm$0.1 | 57.8 $\pm$1.3 | 93.6 $\pm$0.6 | 83.3 |
| MANAR-16K.128.128[†] | 84.0/**84.6** | 91.2 | 71.7 $\pm$0.9 | **90.8** $\pm$0.8 | 90.2 | 90.0 $\pm$0.2 | 61.7 $\pm$0.4 | 94.1 $\pm$1.0 | **84.3** |

MNLI is reported as matched/mismatched (m/mm). Best results are in **bold**; second-best are underlined. [†] Models where every third layer uses a memory size of 16K; all other layers use a memory size of 484, and an ACR size of 64. [‡] Reference rows (BERT, data2vec) are taken as reported and are not retrained. RoBERTa and the MANAR rows are controlled runs trained from scratch by us under one identical recipe (same data subset, optimizer, schedule, and evaluation), so the apples-to-apples comparison is RoBERTa vs. MANAR. MANAR averages are computed over the eight tasks using MNLI-matched. The MANAR-484 → MANAR-16K average improves from 83.3 to 84.3.

leaving others within seed variance. We report mean $\pm$ std over 3–5 seeds for the tasks where multiple seeds were run.

Finally, while newer encoders (Warner et al., 2024; Weller et al., 2025) can reach higher absolute GLUE numbers, they typically differ from our setup by adopting RoPE, longer-context training regimes, and much larger pre-training data. Because our goal here is to attribute changes in downstream quality specifically to the contextualization mechanism, we keep these controlled GLUE comparisons on absolute positional embeddings; MANAR's RoPE re-parameterization is itself implemented and validated in the long-context transfer setting (Sec. 4.4), while a full GLUE sweep under RoPE and longer-context pre-training remain future work.

## 4.2 Image Classification

Furthermore, we benchmark MANAR on the ImageNet-1K dataset (Deng et al., 2009), which contains 1.28M training images and 50K validation images from 1000 categories. We use DeiT-B and DeiT-S (Touvron et al., 2021) as our baselines, hence, we refer to MANAR-M.m.C-B(-S) as a DeiT-B(-S) transformer encoder where all MHA layers were replaced by MANAR layers. We trained a MANAR-256.32.96-B(-S) model with 12 encoder layers, each containing 12(6) heads. The dimensionality of each head was set to $d = 64$.

As in all experiments, only the attention blocks are replaced (MHA → MANAR ); the backbone, augmentation, optimizer, and schedule remain unchanged. Timing and memory comparisons are reported in Sec. 4.7. The model is trained on the training set and the top-1 accuracy on the validation set is reported. For fair comparisons, we trained the model from scratch with training settings used in DeiT. Specifically, we apply random cropping, random horizontal flipping, label smoothing regularization, mixup, and random erasing as data augmentations. The training took place on images of size $224^2$. We employ AdamW (Loshchilov et al., 2017) with $\beta_1 = 0.9$, a total batch size of 1024, and a weight decay of $5 \cdot 10^{-2}$ to optimize the model. We train the MANAR -based DeiT architecture for 450(300) epochs using the cosine scheduling with a learning rate initiated as $4 \cdot 10^{-4}$ and Exponential Moving Average (EMA). During testing we apply a center crop on the validation set to crop out $224^2$ images. Experiments are performed on a single H100 GPU. Top-1 accuracy on validation set results are reported in Tab. 2. We compare against DeiT-B(-S) (Touvron et al., 2021), Vision Mamba Vim-B(-S) (Zhu et al., 2024), a linear-complexity architecture, and the vanilla Vision Transformer (Dosovitskiy et al., 2020). As Table 2 shows, MANAR is competitive with models of comparable size while operating at linear complexity (under fixed local context window).

Table 2: Comparison of different backbone architectures on ImageNet-1K.

| Model | Image Size | Top-1 Acc. (%) |
|---|---|---|
| **Small Models** | | |
| DeiT-S (Touvron et al., 2021) | $224^2$ | 79.8 |
| Vim-S (Zhu et al., 2024) | $224^2$ | 80.3 |
| `MANAR-256.32.96-S` | $224^2$ | 80.7 |
| `MANAR-4K.32.96-S`[†] | $224^2$ | **81.6** |
| **Base Models** | | |
| ViT-B/16 (Dosovitskiy et al., 2020) | $384^2$ | 77.9 |
| ViT-L/16 | $384^2$ | 76.5 |
| DeiT-B | $224^2$ | 81.8 |
| Vim-B (Zhu et al., 2024) | $224^2$ | 81.9 |
| `MANAR-256.32.96-B` | $224^2$ | 82.3 |
| `MANAR-4K.128.96-B`[†] | $224^2$ | **83.9** |

[†] Models where every third layer uses a memory size of 4K; all other layers use a memory size of 256.

## 4.3 Knowledge Transfer

A defining property of MANAR is that it serves as a compatible re-parameterization of MHA, exposing `q`, `k`, `v`, and `out_proj` projection matrices with the same semantic roles as in standard attention. As discussed in Sec. 1, alternative linear-time architectures such as Mamba (Gu & Dao, 2023) and RetNet (Sun et al., 2023) adopt structurally different recurrent or state-space formulations and therefore cannot directly inherit pretrained Transformer attention weights, while latent-bottleneck architectures such as Perceiver IO (Jaegle et al., 2022) introduce a latent array that breaks the per-token Q/K/V parameterization. Among these alternatives, MANAR's preservation of MHA's parameterization makes weight-copy initialization from a standard pretrained Transformer straightforward.

Concretely, knowledge transfer from a pretrained transformer to MANAR is performed by copying the `q`, `k`, `v`, and `out_proj` matrices from each MHA layer into the corresponding MANAR layer. During the initial transfer phase, these copied weights are frozen, and only the newly introduced memory-related parameters are trained. This procedure preserves the inductive biases and representational structure learned by the original model, while allowing MANAR to augment contextualization through its external memory.

We evaluate this transfer mechanism on three representative domains: language understanding, image classification, and automatic speech recognition.

For language understanding, we start from a pretrained RoBERTa model that achieves 83.4% average GLUE after fine-tuning. All MHA layers are replaced with MANAR layers (`MANAR-484.64.128`, i.e., $M=484$, $m=64$, $C=128$) initialized by copying the RoBERTa attention weights. The model is then trained for 5K MLM pre-training steps: the copied weights are frozen for the first 3K steps, after which all weights are rendered trainable for the remaining 2K steps. After fine-tuning, this transferred model achieves 83.5% average GLUE, slightly exceeding the source model while requiring only a fraction of the 30K-step from-scratch training budget used in Sec. 4.1.

For image classification, we start from a pretrained DeiT model achieving 83.4% top-1 accuracy on ImageNet-1K. All MHA layers are replaced with MANAR layers initialized by copying the DeiT attention weights. We evaluate three transfer configurations, all reported as MANAR rows in Table 2. (i) In the *frozen* configuration, each MANAR layer uses memory size $M=256$, ACR size $m=32$, and context window $C=96$, with all copied weights frozen and only the newly introduced parameters trained for 20 epochs; this yields 83.1% top-1 accuracy while updating only a small fraction of the model parameters. (ii) In the *partial-unfreeze* configuration, the small-memory model is trained for 50 epochs total — 20 epochs frozen followed by 30 epochs with all weights trainable — yielding 83.7% top-1 accuracy. (iii) In the *full hybrid* configuration (`MANAR-4K.128.96-B`[†] in Table 2), every third layer uses an enlarged memory of $M=4K$ with $m=128$ while all other layers retain the small-memory setting; following the same partial-unfreeze schedule, this achieves

Table 3: Word Error Rate (WER; %) on LibriSpeech standard dev/test sets. All models use the same 12-layer Transformer encoder. Decoding uses the official 4-gram language model (Heafield, 2011). Lower is better.

| Model | dev-clean | dev-other | test-clean | test-other |
|---|---|---|---|---|
| wav2vec2.0 (Baevski et al., 2020) | 2.7 | 7.9 | 3.4 | 8.0 |
| HuBERT (Hsu et al., 2021) | 2.7 | 7.8 | 3.4 | 8.1 |
| data2vec (Baevski et al., 2022) | 2.2 | **6.4** | 2.8 | 6.8 |
| `MANAR-256.64.128` | 2.3 | 6.7 | 2.9 | 6.8 |
| `MANAR-4K.128.128`[†] | **2.0** | **6.4** | **2.7** | **6.4** |

Best results are in **bold**; second-best are underlined. [†] Models where every third layer uses a memory size of 4K; all other layers use a memory size of 256.

83.9% top-1 accuracy, surpassing the original DeiT baseline by 0.5 pp. Compared with training MANAR from scratch for 450 epochs, all three configurations represent substantial reductions in training cost.

A parallel study is conducted for automatic speech recognition using data2vec-base as the source model. Following the same controlled protocol, we report WER under a consistent decoding setup (including the same language model) across all compared models. As in the vision setting, attention weights are copied into MANAR to initialize the model. A local context window of $C$=128 is employed, corresponding to approximately 2.5 seconds of audio. The model is trained on 100 hours of LibriSpeech using the CTC loss, with all weights rendered trainable during training. In the baseline configuration with memory size $M$=256 and ACR size $m$=64, MANAR matches strong self-supervised speech baselines. Increasing the memory capacity to $M$=4K with $m$=128 in every third layer yields consistent improvements across all evaluation splits, achieving 2.7% / 6.4% WER on the LibriSpeech test-clean / test-other sets (Table 3), competitive with the strongest published baselines.

Taken together, these results indicate that MANAR supports knowledge transfer from pretrained transformers across modalities. Unlike architectures that replace attention with fundamentally different mechanisms, MANAR preserves attention's parameterization while extending it with an expandable memory. This compatibility enables rapid adaptation, substantial reductions in training cost, and continued performance gains as memory capacity grows.

### 4.4  Long-context evaluation

The efficiency advantage of MANAR materializes at long sequence lengths, so we evaluate downstream *quality* there rather than relying on single-layer latency curves alone. We transfer from strong pretrained long-context sources and fine-tune on genuinely long inputs (Table 4). On arXiv long-document classification (documents averaging ∼9,800 words), MANAR transferred from Longformer-base-4096 reaches 88.1% test accuracy at a 4,096-token context, essentially matching the full Longformer (88.3%) while using linear-time attention. Transferring from the *full-attention, RoPE* encoder ModernBERT-base and fine-tuning at 8,192 tokens, MANAR reaches 82.5% versus 82.9% for the original full-attention model under an identical matched protocol — within half a point at linear cost. At high resolution ($384^2$, 577-token sequences), a frozen transfer from DeiT-III-B reaches 85.1% top-1, slightly exceeding the source model (84.8%). The 8,192-token run also exercises MANAR 's RoPE re-parameterization.

**Rotary position embeddings.**  MANAR is compatible with RoPE. Because RoPE encodes *relative* sequence position, we apply it only where relative position is the operative geometry: the rotation is applied to the token query/key that drive the local window, while the retrieved-memory ACR is left position-agnostic (its keys are anchored at position 0), and the retrieval / ACR-construction paths — which are content-based — are not rotated. This selective application requires no change to the contextualization kernel and lets MANAR inherit the weights of RoPE-based encoders such as ModernBERT directly, with each layer using its source layer's RoPE base.

Table 4: Long-context downstream *quality* via weight transfer. Each MANAR model inherits the $q/k/v/\texttt{out}$ projections (and embeddings/FFN/norms) of a strong pretrained source and replaces attention with linear-time local-window+ACR; we then fine-tune. For each task we also list the source/full-attention model under a matched protocol as the baseline ("ref."). Vision context is the patch-token count at $384^2$; text context is in tokens. Sources: DeiT-III (Touvron et al., 2021), Longformer (Beltagy et al., 2020), ModernBERT (Warner et al., 2024).

| Task | Model | Attn | Context | Acc. |
|---|---|---|---|---|
| ImageNet-1K ($384^2$) | DeiT-III-B/16 (ref.) | full | 577 | 84.8 |
| | $\rightarrow$ MANAR | local+ACR | 577 | 85.1 |
| arXiv doc-cls | Longformer-base-4096 (ref.) | full | 4,096 | 88.3 |
| | $\rightarrow$ MANAR | local+ACR | 4,096 | 88.1 |
| arXiv doc-cls | ModernBERT-base (ref.) | full, RoPE | 8,192 | 82.9 |
| | $\rightarrow$ MANAR | local+ACR, RoPE | 8,192 | 82.5 |

arXiv-classification is the 11-class long-document benchmark (documents average $\sim$9,800 words). The 4,096 number is held-out test accuracy; the 8,192 row is a matched-protocol head-to-head against the *original full-attention* ModernBERT (same data/recipe), where MANAR is within $\sim$1 point at linear cost using the RoPE re-parameterization. *Training cost (single L40S):* the long-context fine-tunes are inexpensive — the 4,096 run is $\sim$0.9 h/epoch ($\sim$8 GB) and the 8,192 run $\sim$1.0–1.7 h/epoch ($\sim$32 GB).

## 4.5 Retrieval ablations and weight-transfer controls

To verify that MANAR 's gains come from *meaningful retrieval* rather than the inherited weights or the added parameters, we run controls in which every variant shares the same transferred $q/k/v/\texttt{out}$ weights (Table 5). Shuffling the retrieved concepts across the batch — keeping all parameters but breaking the input-conditioned correspondence — is the most damaging change ($-3.6$), indicating the memory performs genuine input-conditioned retrieval, not generic parameter storage. A frozen-random memory bank degrades accuracy ($-1.3$), removing the ACR memory entirely and keeping only the local window (with the same inherited weights) costs $-2.8$, and MANAR exceeds standard full attention built from the identical inherited weights ($+1.7$). The improvement is therefore attributable to the learned memory pathway and not to parameter count or training budget. Full parameter, trainable-parameter, and memory-specific counts for all variants are given in Table 6; the added capacity is dominated by the search-pattern projection and concept-key map rather than the memory bank itself ($<$1M parameters), which is precisely why these parameter- and compute-matched controls are needed. Varying the retrieval *mechanism* while preserving input-conditioning leaves accuracy essentially unchanged — nearest-neighbour retrieval without product keys (83.4), value-only (83.6), and key-only (82.6) all stay within $\sim$1 point of MANAR — so the product-key factorization is an efficiency choice for scaling the memory, not the source of the gains. Conversely, replacing input-conditioned retrieval with a fixed bank of input-independent learned tokens (a Perceiver-latent-style control) drops accuracy to 80.0 ($-2.9$), close to disabling the memory altogether, confirming that the input-conditioned retrieval — not merely the presence of a global bottleneck — is what matters. Finally, parameter-efficient transfer methods (LoRA, adapters) are orthogonal to MANAR: they leave the attention's quadratic complexity unchanged and so do not provide its linear-time long-context computation; the same-weights full-attention control above (81.2) upper-bounds what such a method on full attention could reach, which MANAR exceeds at linear cost. Architectural ablations over the local window, ACR size, and memory capacity are reported separately in Sec. 4.8.

## 4.6 Analysis of the learned memory

We probe the learned memory of the trained arXiv-4096 model directly (Table 7). The bank is broadly utilized — on average 61.8% of cells are retrieved, with a per-document retrieval entropy of 6.2 of a possible 8.0 bits — so the memory does not collapse onto a handful of cells. Different heads retrieve nearly disjoint

Table 5: Retrieval ablations and same-weights controls on arXiv-4096 (Longformer→MANAR transfer; 1 epoch each, identical data/recipe). All rows share the *same* inherited $q/k/v/$`out` weights, so differences isolate the contribution of the memory/retrieval pathway — not the transferred weights, parameter count, or training budget.

| Variant | What changes | Acc. |
|---|---|---|
| **MANAR (full)** | learned product-key local+ACR retrieval | **82.9** |
| *Same-weights / parameter-matched controls (I2)* | | |
| Full attention | standard MHA, same weights, no ACR | 81.2 |
| Local-window only | no ACR memory at all | 80.1 |
| Frozen-random memory | memory bank fixed at random init | 81.6 |
| *Retrieval-mechanism ablations, input-conditioning preserved (I3)* | | |
| NN retrieval (no product keys) | exact top-$k$ over full-dim cell keys | 83.4 |
| Value-only memory | zero retrieved query/key (values only) | 83.6 |
| Key-only memory | zero retrieved values (addressing only) | 82.6 |
| *Breaking input-conditioned retrieval (I3)* | | |
| Fixed memory tokens | input-independent learned latents (no retrieval) | 80.0 |
| Shuffled memory | break input↔memory correspondence | 79.3 |

Variants that *preserve* input-conditioned retrieval while changing its mechanism — nearest-neighbour retrieval without product keys, value-only, and key-only — all stay within ∼1 point of full MANAR, so the product-key factorization is an efficiency choice for scaling the memory, not the source of accuracy. In contrast, every variant that *breaks* input-conditioned retrieval degrades sharply: replacing retrieval with a fixed bank of input-independent learned tokens (Perceiver-latent style) costs −2.9, shuffling the retrieved concepts across the batch (keeping every parameter) costs −3.6, and disabling the memory entirely (local-window only) costs −2.8. MANAR therefore performs meaningful, input-conditioned retrieval rather than acting as a generic parameter bank, and its gains are not attributable to the transferred weights or added parameters — it exceeds same-weights full attention by +1.7.

Table 6: Parameter accounting for MANAR variants. "MHA base" is the unmodified backbone; "MANAR total" is after replacing every MHA layer. "Mem-only trainable" is the parameter count optimized in the frozen knowledge-transfer setting (copied $q/k/v/$`out` frozen); "memory bank" is the external concept store alone. Param counts are exact; per-run wall-clock and peak memory for the long-context fine-tunes are reported in Table 4.

| Config | MHA base | MANAR total | Mem-only trainable | Memory bank |
|---|---|---|---|---|
| DeiT-S, `MANAR-256.32.96` | 22.05M | 28.16M | 6.11M (21.7%) | 0.75M |
| DeiT-B, `MANAR-256.32.96` | 86.57M | 108.75M | 22.18M (20.4%) | 0.90M |

The added capacity is dominated by the per-layer search-pattern projection and concept-key map, not by the memory bank itself (<1M). Because MANAR adds parameters over its MHA base, we report parameter- and compute-matched controls (local-window-from-MHA; MHA trained for the same extra budget) so that gains are not attributable to parameter count alone.

sets of cells (mean pairwise Jaccard 0.02), indicating strong head specialization, and retrieval is stable to input perturbation. We define *retrieval stability* as the Jaccard overlap between the set of memory cells retrieved from a document and the set retrieved after randomly dropping 15% of that document's tokens; MANAR attains a stability of 0.93 (i.e. 93% of the retrieved cell set is unchanged), evidence that retrieval is a robust, input-conditioned function rather than noise. Across depth, later layers recruit more of the bank (coverage rising to ∼89%) and become slightly less stable, mirroring the late-layer increase in out-of-hull behaviour (Sec. 4.9) and consistent with renewed memory use when the model synthesizes higher-level, task-specific abstractions.

Table 7: Analysis of MANAR's learned memory on the trained arXiv-4096 model ($M$=256 cells/layer, 64 validation documents). *Coverage* is the fraction of memory cells ever retrieved; *entropy* is the per-document retrieval entropy in bits (uniform use of all 256 cells =8.0); *head overlap* is the mean pairwise Jaccard of the cell sets retrieved by different heads (lower = more specialized); *stability* is the Jaccard overlap of retrieved cell sets when 15% of input tokens are randomly dropped (higher = more robust). Reported as the mean over the 12 encoder layers, with the trend across depth.

| Property | Mean (12 layers) | Trend across depth |
| --- | --- | --- |
| Cell coverage | 61.8% | rises 68→89% in late layers |
| Retrieval entropy (bits) | 6.21 / 8.0 | broad, mildly peaked |
| Cross-head overlap (Jaccard) | 0.023 | near-disjoint at all depths |
| Retrieval stability | 0.93 | 0.96 early → 0.88 late |

The memory is broadly utilized (most cells are retrieved; entropy well above the degenerate regime), heads are strongly specialized (near-disjoint retrieved sets, ∼2% overlap), and retrieval is stable to input perturbation (93% overlap under 15% token dropout) — consistent with meaningful, input- conditioned retrieval rather than a generic parameter bank. Late layers recruit a larger and slightly less stable set of cells, matching the late-layer rise in out-of-hull behaviour in Sec. 4.9 and indicating renewed memory use when synthesizing task-specific abstractions.

### 4.7 Performance Comparison

We profile MANAR on a single NVIDIA H100, reporting wall-clock latency averaged over repeated runs and peak allocated memory during inference. Throughout, MANAR uses 256 memory cells, an ACR of 32, and a fixed local context window $C$=128 — the *strict-linear regime*, in which each token attends to a constant-size local window plus the constant-size ACR, giving $O(n)$ time and memory. (A conservative variant sets `cw_len` = $n/2$, which is asymptotically $O(n^2)$ with a much smaller constant than full attention; we use the fixed-window setting for all comparisons below.)

**Measurement protocol.** All timings are wall-clock latencies measured on a single H100 with explicit CUDA synchronization around each timed region; we discard warmup iterations (to absorb one-time kernel autotuning and caching) and report the mean over repeated timed runs. Peak memory is the peak allocated by the CUDA caching allocator during the timed region. The single-layer comparison against standard MHA (Fig. 3) runs in fp32 against a non-FlashAttention MHA layer; the efficient-attention comparison (Fig. 4, Table 8) runs in bf16 at matched $D$=768, 12 heads, $d$=64, and parameter budget, with MHA using a FlashAttention-2 kernel. For each sequence length the batch size is set to the largest that fits in HBM (batch 8 for the matched-setting Table 8). Baselines (Linear Attention, RetNet in its parallel form, Perceiver IO, Mamba-2) are faithful reference implementations chosen to reflect each mechanism's scaling shape rather than vendor-tuned kernels; Mamba-2 uses its Triton selective-scan kernel on the non-fused path, a mild upper bound on its latency.

**Cost of memory retrieval.** We isolate the cost of the retrieval pathway on a single MANAR layer ($D$=768, 12 heads, $M$=256, $m$=32, $C$=128, batch 8, fp32, CUDA-synchronized, 10 warmup / 30 measured runs). The product-key search itself is *constant-time* — ≈0.31 ms independent of sequence length, as expected for top-$k$ over fixed-size factorized key tables — so its share of the layer *falls* from 11% at $n$=512 to 2.0% at $n$=4,096; it is not the bottleneck at the long contexts where MANAR is used. The remaining retrieval cost (the integration phase, in which the $m$=32 ACR slots attend over the input tokens) scales *linearly* in $n$ like the rest of the layer, preserving the overall $O(n)$ complexity. The memory keys are trained jointly end-to-end with the rest of the model; there is no separate retrieval-training stage.

**Versus standard MHA.** Figure 3 compares a single MANAR layer against a standard (non-FlashAttention) MHA layer as the sequence length grows. MANAR's latency and memory grow linearly in $n$, whereas MHA grows quadratically: at $n$=4,096 MANAR already uses 11× less memory (and is several times faster), and at $n$=8,192 the memory gap reaches 23× with a latency gap of roughly 15×; beyond this

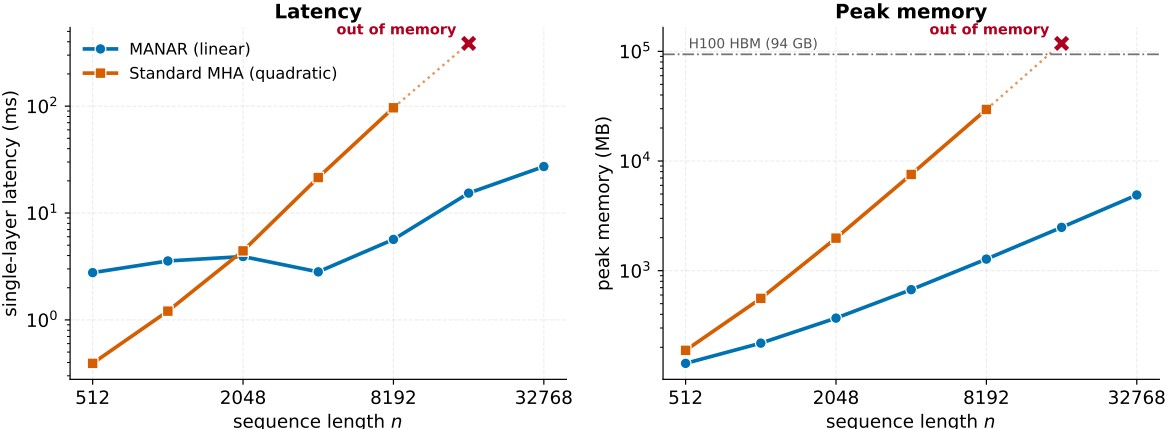

Figure 3: Single-layer latency (left) and peak memory (right) vs. sequence length $n$, MANAR (fixed window $C$=128) against a standard non-FlashAttention MHA layer at matched $D$=768 (one H100, log–log axes). MANAR scales linearly in both and runs through $n$=32K; MHA scales quadratically and exhausts GPU memory beyond $n$=8K (red × marks the first out-of-memory point, where the quadratic extrapolation, dotted, exceeds the H100 HBM capacity line).

point standard MHA exhausts GPU memory while MANAR continues to scale. Speedup and memory figures are insensitive to the ACR size (16–512) and memory size (256–16K). A linear fit of MANAR's fixed-window single-layer latency against $n$ is near-perfect ($R^2$=0.998), confirming the strict-linear regime visible in Fig. 3. In an end-to-end vision model, replacing all MHA layers in DeiT-S with MANAR (`MANAR-256.32.96`-S) yields a 2.0× speed-up and 4.5× memory reduction at 896×896 inputs, rising to 3.1× faster and 8.2× leaner at 1,280×1,280.

**Versus efficient attention alternatives.** Figure 4 compares MANAR against FlashAttention-2 (Dao, 2023), Mamba-2 (Dao & Gu, 2024), RetNet (Sun et al., 2023), Linear Attention, and Perceiver IO (Jaegle et al., 2022) in bf16 at matched $D$=768 and parameter budget. The picture is nuanced and we report it plainly: FlashAttention-2 and the leanest linear layers (Linear Attention, Perceiver IO) are faster than MANAR at short-to-medium lengths, where MANAR's retrieval/ACR overhead dominates. MANAR's advantage is a long-context one — it overtakes FlashAttention-2 at $n \gtrsim 8K$ (about 2.3× faster at 16K, growing to 5.5× at 32K, where it remains well under FlashAttention-2's quadratic compute) and is essentially tied with Mamba-2 at long $n$; RetNet in its parallel form is $O(n^2)$ in memory (like MHA) and exhausts GPU memory beyond $n$=8K, though its chunkwise-recurrent form is $O(n)$ at the cost of sequential execution. We therefore do not claim MANAR is the fastest sub-quadratic layer. Its distinguishing property is that, alone among these mechanisms, it is a re-parameterization of MHA and inherits pretrained attention weights by direct copy (Sec. 4.3); on accuracy it matches or exceeds Vision Mamba (Table 2). In short, MANAR trades a constant-factor overhead at short sequences for weight transferability and linear long-context scaling. Mamba-2 is measured with its Triton selective-scan kernel on the non-fused path (a mild upper bound on its latency), and MLA remains an $O(n^2)$ within-attention KV reduction.

## 4.8 Ablation Studies

Complementing the retrieval-meaningfulness ablations and weight-transfer controls of Sec. 4.5 (which isolate *whether* the memory is used), here we study *how* MANAR 's architectural hyperparameters trade off accuracy and efficiency. Using the DeiT-S configuration from Sec. 4.2, we conduct ablation studies to analyze the impact of MANAR 's key architectural components, focusing on the interplay between local contextualization, the Abstract Conceptual Representation (ACR), and the capacity of the memory used for retrieval.

We study the effect of varying the local context window length (CWL) together with the ACR size. MANAR contextualizes each token using a combination of local neighborhood information and a global ACR

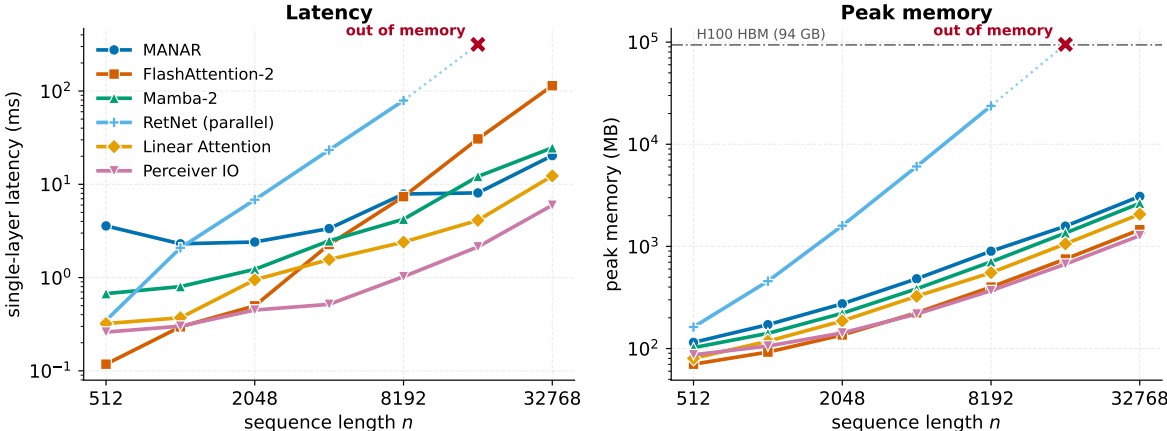

Figure 4: Half-precision (bf16) single-layer latency (left) and peak memory (right) vs. sequence length $n$ at matched $D$=768, for MANAR and the main efficient-attention alternatives (FlashAttention-2, Mamba-2, RetNet, Linear Attention, Perceiver IO). FlashAttention-2 and the leanest linear layers lead at short/medium $n$; MANAR overtakes FlashAttention-2 in the long-context regime ($n \gtrsim 8K$) and ties Mamba-2. RetNet's parallel form is $O(n^2)$ and runs out of memory beyond $n$=8K (red ×). MANAR's differentiator is pretrained-weight transfer (Sec. 4.3), not raw speed.

Table 8: Matched-setting single-layer efficiency vs. linear-time baselines (batch 8, one H100, $D$=768, 12 heads, $d$=64, parameter-budget matched). Latency in ms / peak memory in MB. MANAR uses the strict-linear regime (`cw=128`). RetNet is the parallel form. MHA uses a FlashAttention kernel. "–" denotes out-of-memory.

| $n$ | Linear Attn | RetNet | Perceiver IO | MANAR | MHA |
|---|---|---|---|---|---|
| 512 | 1.9 / 159 | 2.3 / 310 | 1.7 / 134 | 3.2 / 207 | 2.2 / 120 |
| 1024 | 3.0 / 273 | 6.6 / 1012 | 4.0 / 197 | 7.9 / 358 | 4.9 / 195 |
| 2048 | 6.3 / 499 | 16.7 / 3717 | 6.9 / 322 | 8.6 / 660 | 11.1 / 346 |
| 4096 | 10.2 / 953 | 63.2 / 14337 | 12.7 / 574 | 21.6 / 1264 | 39.7 / 648 |
| 8192 | 22.3 / 1861 | 220.1 / 56415 | 23.6 / 1077 | 39.6 / 2472 | 147.0 / 1252 |

Each cell is latency / memory. MANAR scales linearly and beats MHA at long $n$; it is not the leanest linear layer, but it is the only one of these that inherits pretrained MHA weights by direct copy (Sec. 4.3). Baselines are faithful reference implementations for scaling shape, not vendor-optimized kernels.

constructed from retrieved memory concepts. The results show that reducing the context window degrades accuracy significantly only when the window becomes extremely small (e.g., 25% of the sequence), indicating that purely local contextualization is insufficient in that regime. Importantly, increasing the ACR size consistently mitigates this degradation: even with reduced local context, a moderately sized ACR enables MANAR to recover most of the lost accuracy. This suggests that the retrieved global representation provides a strong substitute for long-range token interactions, reducing the reliance on full all-to-all contextualization.

We next examine the impact of memory size in conjunction with ACR size. In contrast to the local context window, memory capacity exhibits a consistent trend: as memory size increases, accuracy improves steadily across all ACR sizes, with no indication of saturation up to the largest bank we evaluate ($M$=16K). The consistent gain across vision (here), language (Table 1), and speech (Table 3) indicates the effect is not specific to a single backbone or modality. Larger memory enables MANAR to retrieve a more diverse and informative set of concepts, which in turn leads to a more expressive ACR. While small memory severely limits performance, enlarging the memory bank continues to yield measurable gains, highlighting memory capacity as a primary driver of model accuracy in our setting.

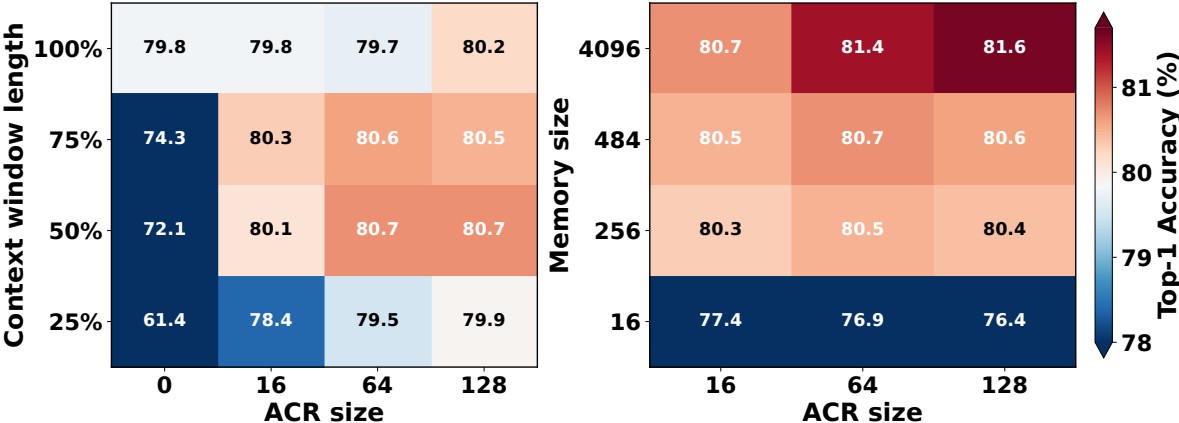

Figure 5: Ablation over local context window length and ACR size (left), and over memory size and ACR size (right). Colormap is normalized to [78, 81.7] to highlight differences in the high-accuracy regime; values outside the range saturate.

Taken together, these ablation studies show that MANAR benefits from increased memory capacity, with accuracy improving consistently as more memorized concepts become available for retrieval. While local contextualization and ACR size control how retrieved information is integrated, memory size determines the richness of the global representation that guides contextualization. This consistent within-range accuracy–capacity relationship supports the central design premise of MANAR : augmenting attention with a sufficiently large, retrievable memory enables strong performance without resorting to quadratic all-to-all interactions, providing a useful trade-off between accuracy and efficiency.

### 4.9 Measuring non-convex contextualization

To quantify MANAR 's ability to produce contextualized representations that are not expressible as convex combinations of attended value vectors, we adopt the Convex Hull Membership (CHM) criterion. Given a contextualized output vector $y_i \in \mathbb{R}^d$ and the set of value vectors $\{v_1, \ldots, v_n\}$ involved in its contextualization, CHM tests whether $y_i$ can be expressed as a convex combination of these values. If no such combination exists, the representation lies outside the convex hull of the inputs.

We emphasize that CHM is a *geometric* diagnostic of representational expansion beyond convex attention-style aggregation; we do not interpret it as a measure of human-like creativity or reasoning. Because standard softmax MHA outputs are by construction convex combinations of the input values, the CHM rate of MHA is identically zero; any non-zero CHM rate is attributable to the additional value sources introduced by MANAR (the retrieved memory concepts).

To ensure that CHM measurements reflect an intrinsic property of the MANAR layer—rather than a consequence of retraining or task-specific optimization—we evaluate CHM under a strictly controlled setting in which all transformer-based weights are frozen. Specifically, MANAR is initialized via knowledge transfer from pretrained transformer encoders, and only the additional parameters introduced by MANAR (i.e., memory retrieval, ACR construction, and memory-to-token projection weights) are trained. For language modeling, weights are transferred from `bert-base-uncased`. For image classification and speech recognition, we use the intermediate checkpoints from the knowledge-transfer experiments in Sec. 4.3, prior to unfreezing the copied weights.

This design isolates the architectural contribution of MANAR itself. By freezing all original transformer parameters, we ensure that any deviation from convex-hull-limited representations cannot be attributed to changes in the underlying attention projections, token embeddings, or feed-forward blocks. Instead, any out-of-hull behavior must arise solely from augmenting a pretrained transformer with MANAR 's memory. In other words, this experiment tests whether expanding MHA into MANAR without altering the learned

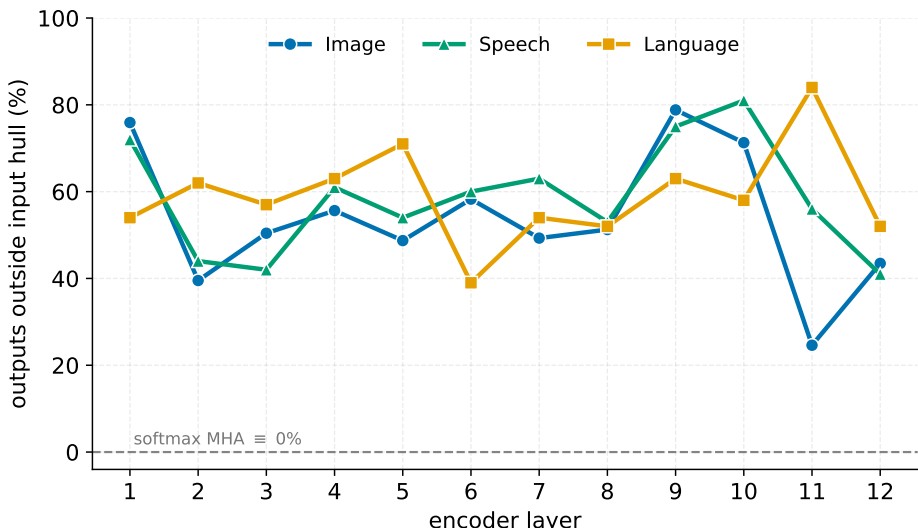

Figure 6: Fraction of layer outputs lying outside the convex hull of input values (CHM) across encoder layers for image, speech, and language models. Standard softmax MHA is identically 0% (dashed line); MANAR's memory pathway lifts a large fraction of outputs out of the input hull at every layer.

transformer representation space enables the model to express representations not reachable by the original convex aggregation.

CHM measurements are conducted on `MANAR-256.32.96-B` (image), `MANAR-256.64.128` (speech), and `MANAR-484.64.128` (language). For each encoder layer, we uniformly sample 10,000 output tokens across all heads and solve the CHM feasibility problem using linear programming. Figure 6 reports the fraction of outputs lying outside the convex hull for each layer and modality.

Across all three modalities, a large fraction of representations, often exceeding 50%, lie outside the convex hull of the input values, indicating that MANAR consistently produces non-convex contextualizations even when built on top of a frozen transformer. Early layers exhibit substantial out-of-hull behavior, reflecting reliance on memory-driven global context when local evidence is limited. Mid-stack layers show a temporary stabilization as token-level features consolidate. In later layers, the out-of-hull fraction rises sharply, indicating renewed use of memory concepts when synthesizing higher-level, task-specific abstractions, before declining near the output layer.

The non-convex behavior observed across modalities reflects the architectural property that representations synthesized via memory-augmented attention can lie outside the input value hull, expanding the space of expressible token representations beyond what convex aggregation over the input alone permits. This property is a direct consequence of MANAR 's memory-augmented design rather than a retraining effect, since the underlying transformer weights are frozen throughout the experiment.

We position CHM as a *mechanistic diagnostic* of where MANAR's expressivity comes from, not as a standalone performance metric, and we are careful not to claim that out-of-hull geometry by itself causes higher accuracy. Its practical relevance is, however, concrete and is established by the rest of the paper: the memory pathway that is solely responsible for the non-convex behavior (the CHM rate of the underlying MHA is identically zero) is the same component whose *capacity* drives the accuracy gains reported in Sec. 4.8 (Fig. 5) and Table 1. We confirm the causal role of the memory pathway with a controlled ablation: holding weights and inputs fixed and disabling the memory pathway (reducing MANAR to local-window attention) yields a CHM rate of *exactly* 0% — every output lies within the convex hull of its windowed input values — whereas re-enabling memory produces strictly out-of-hull outputs. The non-convexity is therefore attributable solely to the retrieved memory concepts, not to the local attention or the projections. In other words, the geometric expansion measured here and the downstream gains measured elsewhere are two readouts of the same

architectural mechanism; CHM lets us verify that the mechanism is active even on a frozen backbone, while the ablation and scaling studies quantify its effect on task accuracy.

## 5 Limitations

We note several limitations of the current MANAR formulation. First, the architecture is presented and evaluated in an encoder-only setting; supporting causal, decoder-style attention requires a different memory-update schedule and is left to future work. Relatedly, our long-context downstream evaluation (Sec. 4.4) uses long-document classification and high-resolution vision; generative long-context suites such as SCROLLS and LongBench target decoder-style question answering and therefore fall outside our encoder scope, and long-audio (e.g. concatenated utterances) is a natural extension of our speech pipeline that we leave to future work. Second, our controlled language-modeling comparisons (Sec. 4.1) use absolute positional embeddings to keep them clean against the baselines; Rotary Positional Embeddings (RoPE) are nonetheless supported and validated separately — we transfer from the RoPE-based full-attention encoder ModernBERT-base and fine-tune at 8,192 tokens (Sec. 4.4) — though we have not yet re-run the full GLUE suite under RoPE. Third, we report GLUE results as mean $\pm$ std over multiple fine-tuning seeds (Table 1); pre-training, however, remains single-seed due to compute, so we do not characterize pre-training-level variance. Fourth, the memory-size scaling study explores up to $M$=16K cells; larger banks (e.g., $M$=256K or product-key tables of size $\sim$1M) are likely required to characterize whether the within-range accuracy-vs-capacity trend in Fig. 5 continues or saturates. Fifth, the strict-linear time and memory complexity holds when the local context window is fixed to a constant — the setting used both by our deployed models (e.g. $C$=96/128) and by the efficiency comparisons in Sec. 4.7, which is why MANAR scales linearly there; allowing the window to grow with the sequence (e.g. `cw_len` $= n/2$) would instead be asymptotically $O(n^2)$, albeit with a much smaller leading constant than full MHA. Finally, the GWT analogy used throughout the paper is an interpretive guide for the architectural choice of routing global information through a constant-size ACR, not a claim that MANAR mechanistically realizes a cognitive theory.

## 6 Conclusion

We introduced **MANAR**, a memory-augmented contextualization layer that re-parameterizes standard multi-head attention while drawing on Global Workspace Theory (GWT) for its high-level architectural design. By routing global information through a constant-sized Abstract Conceptual Representation (ACR) and a retrievable external memory, MANAR achieves strictly linear time and memory complexity when the local context window is fixed — a regime we confirm empirically (Fig. 3) — and substantial speedups over standard attention that grow with sequence length.

Across language, vision, and speech modalities, MANAR is competitive with strong baselines (average GLUE 84.3, ImageNet-1K 83.9% top-1, LibriSpeech 2.7%/6.4% WER) while remaining a compatible re-parameterization of MHA that admits knowledge transfer from pretrained models via direct weight-copy — a property that, among recent linear-time alternatives, makes MANAR particularly easy to adopt in existing transformer ecosystems.

As a diagnostic, a Convex Hull Membership analysis shows that MANAR expresses representations outside the convex hull of input value vectors even when grafted onto frozen pretrained transformers, confirming that the memory pathway is active and expands the representational reach of the layer rather than merely substituting for input-side attention; we treat this as evidence the mechanism is engaged, not as a standalone performance claim.

For practitioners, MANAR can be used as a drop-in replacement with knowledge transfer to reduce training cost; it is relevant to information retrieval, document and text analysis, and multimodal or speech-based systems. Several directions remain for future work, including support for causal/decoder-style attention, a full GLUE sweep under RoPE, longer-context pre-training, and scaling the external memory beyond the 16K cells studied here.

**Reproducibility Statement**

We aim to make all reported results reproducible. Training recipes — data subsets, optimizer, schedule, batch size, sequence length, augmentation, and total step count — are specified explicitly for each modality in Sec. 4.1 (language), Sec. 4.2 (image), and the speech subsection of Sec. 4.3. Configuration of MANAR is fully described by the `MANAR-M.m.C` naming convention used throughout, with the additional hybrid-layer specification footnoted in each table. All experiments were run on a single NVIDIA H100 GPU. The performance comparisons in Sec. 4.7 report wall-clock latencies averaged over repeated runs and peak GPU memory measured during inference; the batch size for each sequence length is set to the maximum that fits in HBM. Code (including training scripts, model definitions, and the CHM evaluation pipeline used in Sec. 4.9) will be released upon acceptance at the anonymous URL referenced in Sec. 1; during review, the URL is redacted to preserve double-blind anonymity.

**Use of generative AI in the writing process**

During the preparation of this manuscript the authors used generative AI and AI-assisted tools for rephrasing of selected text and for identifying and organizing relevant references. All content was reviewed, verified, and edited by the authors, who take full responsibility for the accuracy, integrity, and originality of the manuscript.

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
