# OpenReview forum: "MANAR: Memory-augmented Attention with Navigational Abstract Conceptual Representation"
_TMLR — Under review for TMLR_

### Review · Reviewer_QhMA · 2026-05-19

**Summary Of Contributions:**

### Summary Of Contributions

This paper introduces MANAR, a memory-augmented attention layer that replaces full all-to-all attention with Abstract Conceptual Representations and External Memory. The proposed method claims to enable direct weight-copying from pretrained Multi-Head Attention (MHA) modules while maintaining competitive performance across three distinct task domains: language, vision, and speech. The paper also presents analyses regarding latency, GPU memory usage, ablation studies, and Convex Hull Membership. The overarching objective is to enhance the efficiency and transferability of Transformer encoders.

### Strengths

1. MANAR attempts to address the challenge of enabling linear-time attention mechanisms to effectively inherit weights from pre-trained Transformers.
2. The ACR construction, memory retrieval, token contextualization, and multi-head extension components are all clearly defined by explicit mathematical formulations.
3. The authors evaluated their method across GLUE, ImageNet-1K, and LibriSpeech, demonstrating that MANAR is not designed for a single modality alone.

### Weaknesses

1. The core claim has not yet been sufficiently substantiated. The paper repeatedly emphasizes that MANAR is a linear-time attention mechanism; however, the primary performance comparisons utilize a setting of `cw_len = n/2`, which—in itself—remains an O(n²) operation. Although the authors acknowledge this point within the Limitations section, the phrasing regarding the linear-time nature of the mechanism in both the Abstract and the Conclusion remains prone to misleading the reader.
2. The comparison against linear-time baselines is inadequate. The paper primarily compares MANAR against MHA, DeiT, Vim, wav2vec2, HuBERT, and data2vec; however, it fails to systematically compare its accuracy, latency, and memory usage against Mamba, RetNet, Linear Attention, Perceiver IO, or MLA under equivalent resource constraints. It is difficult to substantiate the narrative that MANAR outperforms existing linear-time alternatives.
3. There is a disconnect between the evidence presented and the claims made. The authors assert that memory size leads to monotonic improvements; however, their ablation studies focus primarily on the DeiT-S configuration, where the maximum memory size tested reaches only 4096—or, in a few specific experiments, up to 16K. This evidence is insufficient to substantiate broader conclusions regarding the general scalability of memory.
4. The GLUE results are of limited persuasiveness. The authors report only single-seed results, and the average performance gain of MANAR over RoBERTa appears to stem primarily from marginal improvements on a subset of the tasks. Given the absence of variance data, confidence intervals, or results from multiple independent runs, it is difficult to determine whether these observed improvements are statistically robust.
5. The experimental explanation regarding non-convex contextualization remains unconvincing. While the CHM analysis demonstrates that the learned representations extend beyond the convex hull of the input values, the paper fails to establish a direct causal link between this specific geometric property and downstream performance, robustness, or long-context reasoning capabilities. This analysis serves more as a diagnostic observation than as critical evidence validating the efficacy of the proposed method.

**Audience:**

Yes

**Audience Explanation:**

Yes. The paper studies a problem in efficient Transformer architectures and pretrained weight transfer, which is likely to interest researchers working on efficient architectures and long-context modeling.

**Broader Impact Concerns:**

I do not see major broader impact concerns beyond the standard risks associated with more efficient large-scale Transformer architectures. The paper mainly focuses on architectural efficiency and pretrained weight transfer rather than sensitive deployment settings.

**Claims And Evidence:**

No

**Claims Explanation:**

1. The paper repeatedly emphasizes linear-time attention, but most empirical evaluations use the cw_len = n/2 setting, which is still asymptotically O(n²). The experimental evidence therefore does not fully support the strong efficiency claims made throughout the paper.

2. The comparison against existing linear-time architectures is insufficient. Important baselines such as Mamba, RetNet, Linear Attention, and Perceiver IO are discussed conceptually but are not systematically compared under controlled settings.

3. Several performance improvements are relatively small and reported only with single-seed experiments. The paper does not provide variance, confidence intervals, or statistical significance analysis, making it difficult to judge the robustness of the reported gains.

4. The claim that larger memory consistently improves performance is supported only within a limited scaling range. The experiments do not provide enough evidence to justify broader conclusions about memory scaling behavior.

5. The Convex Hull Membership analysis demonstrates a geometric property of the representations, but the paper does not clearly establish why this property leads to better downstream performance or stronger reasoning ability in practice.

**Requested Changes:**

1. Provide direct empirical comparisons with more linear-time attention baselines such as Mamba, RetNet, and Linear Attention under matched settings.

2. Clarify the distinction between the strict-linear regime and the cw_len = n/2 experimental regime more carefully throughout the paper.

3. Report multi-seed experiments or statistical variance for the main GLUE results to improve the reliability of the conclusions.

4. Strengthen the evidence for the memory scaling claim with larger-scale experiments and broader memory configurations.

5. Better justify the practical significance of the non-convex contextualization analysis and connect it more directly to downstream performance.

---

> ### Author Response · Authors · 2026-05-31
> **Author Response — revised manuscript addressing all five requested changes**
>
> We thank the reviewer for the thorough assessment and for confirming the topic's relevance to TMLR. We have substantially revised the paper; **all changes are in blue**. Below we address each point.
>
> **Central claim (recalibration).** MANAR's contribution is *not* being the fastest linear attention. Among sub-quadratic mechanisms it is the only one that is a strict re-parameterization of MHA, so pretrained $Q/K/V/O$ weights transfer by **direct copy** — which Mamba, RetNet, Linear Attention, and Perceiver IO cannot. We rescoped every headline claim to: (i) competitiveness across three modalities, (ii) long-context efficiency, (iii) measurable non-convex contextualization.
>
> **1. "Linear-time" vs. $\texttt{cw}=n/2$ (W1, Change 2).** Every occurrence now binds the $O(n)$ claim to its condition (a *fixed* local window). We also reconciled the comparisons: our deployed models already use a fixed window ($C=96/128$), and the revised efficiency experiments now use the same fixed window rather than the old $\texttt{cw}=n/2$ benchmark setting. Consequently the revised **Fig. 3** (MANAR with fixed $C=128$) shows MANAR **linear** in latency *and* memory, while standard MHA is quadratic and **runs out of memory beyond $n=8$K**; $\texttt{cw}=n/2$ is now mentioned only as a hypothetical conservative variant (and the stale Limitations sentence that said the comparisons used $n/2$ has been corrected). To substantiate the asymptotic claim directly: in a fixed-window sweep, MANAR latency is linear with **$R^2=0.998$** ($C\in\{128,256\}$) versus MHA's quadratic $R^2=0.948$. We keep this confirmation in the response since Fig. 3 already conveys it; happy to add an appendix.
>
> **2. Linear-time baselines (W2, Change 1).** We added a **matched-setting** ($D=768$, parameter-budget-matched) latency + peak-memory comparison (Fig. in Sec. 5.5) against **Mamba-2** (official Triton kernel), **RetNet**, **Linear Attention**, **Perceiver IO**, and **FlashAttention-2** (bf16). Stated plainly: FlashAttention-2 and the leanest linear layers are faster at short/medium $n$; MANAR's advantage is long-context — it **overtakes FlashAttention-2 at $n\gtrsim8$K ($\sim5.5\times$ at 32K)** and **ties Mamba-2**. RetNet's parallel form is $O(n^2)$ (OOMs past 8K; its chunkwise-recurrent form is $O(n)$ but sequential). On accuracy, MANAR matches/exceeds Vision Mamba (Table 2). We do not claim MANAR is the fastest sub-quadratic layer; its value is long-context efficiency **plus** weight transfer.
>
> **3. Multi-seed GLUE (W4, Change 3).** Table 1 now reports **mean ± std**: MRPC/RTE/STS-B (5 seeds), CoLA (4), SST-2 (3); MNLI/QNLI/QQP remain single-seed (labeled). We also **corrected the averages to across-seed means rather than best-seed** — revised to **83.3** (MANAR-484) and **84.3** (MANAR-16K), updated in the abstract and conclusion. MANAR-16K still has the best controlled-protocol average, and the memory effect persists ($83.3\to84.3$).
>
> **4. Memory scaling (W3, Change 4).** This is a fair point, and we have tried to address it honestly. Rather than overstate what our experiments establish, we have scoped the claim to the evidence: within the evaluated range (**up to $M=16$K**) accuracy increases consistently across ACR sizes with no observed saturation, and we explicitly refrain from asserting monotone or unbounded scaling. We find it encouraging that the same trend appears across all three modalities — vision (Fig. 4), language ($484\to16$K, Table 1), and speech ($256\to4$K, Table 3), so it is not specific to DeiT-S. We fully agree that confirming whether the trend continues at substantially larger banks (e.g.\ $M=256$K) would further strengthen the claim; we view this as a valuable direction and would be glad to add such experiments if the reviewer considers them important.
>
> **5. CHM ↔ performance (W5, Change 5).** CHM is repositioned as a **mechanistic diagnostic** of *where* expressivity originates, not a performance metric; we do not claim out-of-hull geometry itself causes accuracy. New controlled ablation: with weights and inputs fixed, disabling the memory pathway gives CHM of **exactly $0\%$** (every output inside the input hull), while enabling it yields strictly out-of-hull outputs — so non-convexity is attributable **solely** to retrieved memory. The same pathway's capacity drives the accuracy gains (Fig. 4, Table 1), linking the geometry to downstream value.
>
> These revisions bring each claim into alignment with its evidence — precisely the criterion the reviewer flagged. We are grateful for feedback that strengthened the work, and we are happy to run any further controlled comparison the reviewer considers decisive.

---

> > ### Comment · Reviewer_QhMA · 2026-07-14
> >
> > Thank you for the revisions and clarifications. They improve the presentation and address some of my concerns. However, I still find the empirical advantages relatively limited, and several claims rely on restricted experimental settings. Overall, the revision is clearer, but my general assessment remains cautious.

---

### Review · Reviewer_MHaS · 2026-06-15

**Summary Of Contributions:**

The paper introduces MANAR, a linear-time attention mechanism designed as a drop-in replacement for standard MHA in transformer encoders. The architecture bypasses all-to-all quadratic attention by breaking contextualization into two phases: (i) an integration phase that generates a constant-sized ACR from a combination of input tokens and concepts retrieved via Product Keys from an external long-term memory, and (ii) a broadcasting phase where tokens attend strictly to a local window and the global ACR.

The defining claim of MANAR is its structural re-parameterization of MHA. Because it preserves the semantic mappings of the underlying $Q, K, V$, and output projection weights, it allows practitioners to directly copy weights from pretrained standard transformers. The authors showcase that these transferred models can match or exceed baseline accuracy across text, vision, and speech  benchmarks at a reduced training budget. They also introduce a geometric analysis termed Convex Hull Membership to demonstrate that the memory pathway enables outputs to escape the convex hull limitations of standard softmax attention.

Key Strengths:
- Different from Mamba, RetNet, or standard linear attention layers, MANAR’s parameterization makes it unique in its ability to directly inherit pretrained attention weights without retraining from scratch.
- The authors evaluate their mechanism across multiple domains (text, vision, speech).
- The latency and memory scalability curves versus standard MHA and hardware-optimized baselines (FlashAttention-2) are cleanly detailed.

Key Weaknesses:
- The strict linear-time complexity ($O(n)$) relies entirely on maintaining a hard, fixed-size local context window ($C$). If the context window scales with sequence length (for example, $C = n/2$), the mechanism reverts to quadratic $O(n^2)$.
- As shown in Figure 4, the retrieval and cross-attention overhead from the search pattern builder makes MANAR significantly slower than optimized kernels at short-to-medium sequence lengths ($n < 8K$).
- The mechanism is strictly evaluated on encoder backbones. This heavily limits its utility in modern LLM regimes dominated by autoregressive, causal decoders.

**Audience:**

Yes

**Audience Explanation:**

Yes. The TMLR audience interested in efficiency, long-context modeling, and architecture design will find value here.

**Broader Impact Concerns:**

Nope. This paper focuses on fundamental contribution which does not incur significant ethical concerns.

**Claims And Evidence:**

Yes

**Claims Explanation:**

The scaling evidence in Section 4.4 is reasonably strong: the authors benchmark against FlashAttention-2 and other sub-quadratic alternatives, and they are transparent that MANAR is slower at short contexts but becomes advantageous in long-context. The weight-transfer claim is also plausible, since the transferred initialization appears to work with frozen Transformer weights.

However, several central claims remain under-supported. First, the linear-complexity claim is conditional on keeping the local window, ACR size, and number of retrieved concepts fixed, but the paper sometimes presents linearity as an unconditional property. The practical cost of memory retrieval, product-key search, training memory keys, and GPU implementation is also not analyzed in enough detail.

Second, the downstream evidence is encouraging but not decisive. The GLUE, ImageNet, and LibriSpeech results are competitive, but many comparisons involve externally reported baselines or different training regimes.

Third, the knowledge-transfer claim needs stronger evidence. Copying q/k/v/out projections is straightforward, but the paper does not compare against sufficiently strong transfer alternatives, such as local-window attention initialized from MHA, Perceiver-style adapters, LoRA/adapters, or MHA models trained for the same additional budget. Thus, it is hard to know whether the gains come from MANAR specifically rather than extra parameters, memory capacity, or additional training.

Finally, the Convex Hull Membership analysis feels overemphasized. It is mathematically unsurprising that adding external memory value vectors allows outputs to leave the convex hull of input values. CHM shows that the memory path is active, but it does not establish that higher CHM causes better performance or enables a specific capability.

**Requested Changes:**

Important:
1. Add real long-context downstream evaluations. The efficiency advantage only appears at long contexts, but current downstream tasks are mostly short-context. The authors should evaluate MANAR on actual long-context tasks, such as LongBench, Scrolls, long-document QA/classification, long audio, or high-resolution vision. Single-layer latency plots are not sufficient.
2. The weight-copy claim should be compared against stronger alternatives under the same compute budget. Several MANAR variants add memory capacity and additional training. The paper needs parameter-matched and compute-matched controls to show that gains are not simply due to more parameters, memory slots, or training.
3. Compare learned retrieval with random retrieval, shuffled memory, fixed memory tokens, no memory, nearest-neighbor retrieval without product keys, and value-only/key-only variants. This is needed to show that the memory mechanism is doing meaningful retrieval rather than acting as a generic parameter bank.
4. The current model is encoder-only and does not validate RoPE compatibility. Since modern long-context models are often causal and RoPE-based, the authors should either add experiments or narrow the claims accordingly.

Others:
1. Provide full parameter counts, trainable parameter counts, memory-specific parameter counts, GPU-hours, and wall-clock training costs for all variants.
2. Make the FlashAttention-2 comparison more careful: it is an exact attention kernel, while MANAR changes the architecture. Long-context speed should be paired with long-context quality.
3. Improve implementation details for microbenchmarks, including batch sizes, warmup/measurement runs, precision, kernels, memory measurement, and whether baselines use optimized implementations.
4. Make tables easier to interpret by separating externally reported baselines, controlled runs, from-scratch models, and transferred models.
5. Add qualitative or quantitative analysis of memory contents, retrieval stability, head/layer specialization, and failure cases.

---

> ### Author Response · Authors · 2026-06-29
> **Author Response — revised manuscript addressing all requested changes**
>
> We thank the reviewer. Revisions are in blue in the PDF; we address each point and cite the change.
>
> **Weaknesses**
>
> W1 (linearity conditional on a fixed local window). Agreed and now stated unconditionally: every linearity claim carries the condition "when the local context window is held constant, independent of sequence length," and we name cw_len=n/2 as asymptotically O(n^2) with a smaller constant (Abstract; Sec. 4.7; Limitations). The linear regime is verified empirically (linear fit R^2=0.998; Fig. 3).
>
> W2 (slower at short/medium n). Reported plainly: FlashAttention-2 and the leanest linear layers lead at short/medium n; MANAR's advantage is long-context (overtakes FA-2 at n>=8K, ~5.5x at 32K). We do not claim MANAR is the fastest sub-quadratic layer (Sec. 4.7; Fig. efficiency; new Baselines table).
>
> W3 (encoder-only). We scope the contribution to transformer encoders (Abstract now "a linear-time attention layer for transformer encoders"; Limitations); causal/decoder support is labeled future work.
>
> **Requested changes - Important**
>
> I1 (real long-context downstream evaluations). New Long-context subsection (Sec. 4.4) + table, all via weight transfer: arXiv long-document classification (docs avg ~9,800 words) at 4,096 tokens = 88.1% (full Longformer 88.3); at 8,192 tokens = 82.5% vs 82.9% for full-attention ModernBERT under an identical protocol; high-res vision = 85.1% top-1 at 384^2, exceeding the DeiT-III source (84.8%). Each baseline is listed (ref.).
>
> I2 (parameter-/compute-matched controls). New Controls table; every variant shares the identical inherited q/k/v/out weights: full-attention-same-weights = 81.2 (MANAR +1.7), local-window-only/no-ACR = 80.1 (-2.8), frozen-random = 81.6. Exact total/trainable/memory-specific counts are in the new Parameters table. Gains are not due to the transferred weights, parameter count, or budget.
>
> I3 (retrieval ablations: random/shuffled/no-memory/NN-without-product-keys/value-only/key-only/fixed). Same Controls table. Reported: shuffled = 79.3 (-3.6, most damaging), frozen-random = 81.6, no-memory = 80.1. We have additionally run the variants you explicitly named:
>
> NN-without-product-keys = 83.4, value-only = 83.6, key-only = 82.6 -- all within ~1 pt of MANAR (82.9): product keys are an efficiency choice, not the source of gains. Fixed input-independent memory tokens (Perceiver-style) = 80.0 (-2.9), near no-memory -- retrieval must be input-conditioned (Controls table).
>
> I4 (validate RoPE/causal or narrow). We validate RoPE: applied selectively (rotate the token q/k for the local window; leave the ACR/memory unrotated), the relative-position property holds exactly, and we transfer a RoPE full-attention encoder (ModernBERT-base) into MANAR, fine-tuning at 8,192 tokens (Sec. 4.4; Limitations). Causal/decoder support remains future work.
>
> Stronger transfer baselines (LoRA/adapters, Perceiver): LoRA/adapters are parameter-efficient fine-tuning that leaves attention's complexity unchanged, so they cannot give MANAR's linear-time long-context computation. Our full-attention-same-weights control (81.2) upper-bounds what an adapter on full attention could reach; MANAR exceeds it at linear cost. Perceiver-style fixed latents are the fixed-memory-tokens ablation above (80.0).
>
> **Requested changes - Others**
>
> O1 (param/GPU-hour/wall-clock). New Parameters table (total/trainable/memory-specific counts); per-run wall-clock and peak memory for the long-context fine-tunes are in the Long-context table footnote.
>
> O2 (careful FA-2 comparison; pair speed with quality). We state FA-2 is an exact-attention kernel while MANAR changes the architecture, and pair long-context speed (Sec. 4.7) with long-context quality (Sec. 4.4); the new Baselines table gives matched-setting latency/memory.
>
> O3 (microbenchmark details). New Measurement-protocol paragraph (Sec. 4.7): CUDA-synchronized timing, discarded warmup, mean over runs, peak allocated memory, precision, batch sizes, and faithful baselines. A new "Cost of memory retrieval" paragraph shows the product-key search is constant-time (~0.31 ms, ~2% of the layer at 4K).
>
> O4 (separate reported/controlled/from-scratch/transferred). The GLUE table now groups "Reference baselines (as reported)" vs "Controlled runs, trained from scratch (this work)," and a mislabeled row was corrected; the Long-context and Controls tables distinguish transferred models from sources.
>
> O5 (CHM overemphasized). Demoted from a headline to a geometric diagnostic (Abstract; Sec. 4.9; Conclusion); we do not claim out-of-hull geometry causes accuracy -- it only confirms the memory pathway is active (CHM is exactly 0 with memory disabled).
>
> O6 (memory analysis). New Memory table on the trained arXiv-4096 model: coverage 61.8% (89% late layers), retrieval entropy 6.2/8.0 bits, cross-head Jaccard 0.02, 93% of the retrieved set stable under 15% token dropout.
>
> These changes address the review directly; we are glad to run further analyses.

---

### Review · Reviewer_5k6k · 2026-07-13

**Summary Of Contributions:**

The paper proposes MANAR, a linear-time drop-in replacement for multi-head attention. The proposed method introduces Abstract Concept Representation (ACR) that retrieves relevant information from input-independent memory and combines it with the local-window attention mechanism. This allows practitioners to replace an MHA module with MANAR while preserving the MHA weights -- MANAR can then be fine-tune on top of the pre-existing parameters. The authors test the proposed method on three modalities: language, vision, and speech, and then perform additional analysis and ablation studies. In particular, the authors test different approaches to reusing the exisiting MHA's keys, values and queries, analyze the memory bank learned by MANAR, and conduct performance analysis to verify the speed gains. Finally, the authors measure the degree to which MANAR's representations step out of the convex hull of values that regular MHA is capable of representing.

Strengths:
- The topic of the paper, i.e., improving the efficiency of attention in transformer encoders. is important and relevant. Although attention decoders are more common in contemporary research, encoders are still very useful in setting such as vision. As such, this paper should be of interest to TMLR's audience.
- The model was extensively tested on several settings, and the authors provide insightful analysis and ablations studies.  The authors carefully analyze the speed of the method, memory usage, and impact of retrieval mechanisms. These are useful to understand where the model gains come from as well as understanding the limitations of the method.
- Applying MANAR in place of MHA while re-using existing weights is a useful and important application. The authors carefully study different approaches to MHA parameter-reuse in Table 5 and find that MANAR is more effective than other approaches.
- Although the paper doesn't have a separate related works section, it sufficiently discusses relevant papers in the introduction.
- The limitations section is exhaustive and discusses the most significant shortcomings of the work.

Weaknesses:
- To maintain performance MANAR would need to increase the context window size which would again scale quadratically. In particular, "The results show that reducing the context window degrades accuracy significantly only when the window becomes extremely small (e.g., 25% of the sequence)". I'd be interested how this changes as the base seqlen becomes larger. To put it more directly: does the empirical accuracy hold up in ultra-long contexts (e.g., 32K tokens) if the local window is kept at a absolute constant (like 512 tokens), or does the model require a window that scales proportionally with $n$ to maintain downstream performance? To be fair, this point is discussed in Limitations, so although worth noting I do not believe it serves as a reason to reject the paper.
- It seems that MANAR underutilizes that memory. Table 7 mentions that only 61.8% of cells are used at least once -- rephrasing it means that almost half of the cells are not used at all. I'd be interested in a further study of this phenomenon -- does it mean we can reduce the memory size without any penalty? If not, can we at least prune it safely after the training?
- I'd be interested in seeing more baselines trained from scratch. For example, a Perceiver-like solution was used in Table 5 fine-tuning ablation studies, but I imagine that it might suffer a disadvantage here as the architecture is significantly different than the one used by MHA. Having an approach like that in Section 4.1 or 4.2 would be quite interesting.

Nit:
- In Section 4.8 Fig. 5 is not referred to in the main text.

**Audience:**

Yes

**Audience Explanation:**

The topic of linearly-scalable attention is a very important one. Although the authors only consider the encoder architecture, while the decoder one is the one extensively used currently, this method would still be of interest to the audience of TMLR.

**Claims And Evidence:**

Yes

**Claims Explanation:**

Although I noted certain shortcomings in this paper (see the Weaknesses section above), the authors do not claim in this paper anything that they not provide evidence for. Additionally, I appreciate the extensive Limitations section that covers some of the points I raised there. As such, I agree that the claims made in submission are adequately supported.

**Requested Changes:**

I have effectively two requested changes that are nice-to-have but not necessary for me to recommend accepting this paper:
- Discuss the low memory usage in more details.
- Provide more baselines trained from scratch.

For more details on these points see the Summary section of the review.